# TaskPrompter: Spatial-Channel Multi-Task Prompting for Dense Scene Understanding

**Hanrong Ye and Dan Xu**
Department of Computer Science and Engineering
The Hong Kong University of Science and Technology (HKUST)
Clear Water Bay, Kowloon, Hong Kong
`{hyeae,danxu}@cse.ust.hk`

## Abstract

Learning effective representations simultaneously from multiple tasks in a unified network framework is a fundamental paradigm for multi-task dense visual scene understanding. This requires joint modeling (i) task-generic and (ii) task-specific representations, and (iii) cross-task representation interactions. Existing works typically model these three perspectives with separately designed structures, using shared network modules for task-generic learning, different modules for task-specific learning, and establishing connections among these components for cross-task interactions. It is barely explored in the literature to model these three perspectives in each network layer in an end-to-end manner, which can not only minimize the effort of carefully designing empirical structures for the three multi-task representation learning objectives, but also greatly improve the representation learning capability of the multi-task network since all the model capacity will be used to optimize the three objectives together. In this paper, we propose TaskPrompter, a novel spatial-channel multi-task prompting transformer framework to achieve this target. Specifically, we design a set of spatial-channel task prompts and learn their spatial- and channel interactions with the shared image tokens in each transformer layer with attention mechanism, as aggregating spatial and channel information is critical for dense prediction tasks. Each task prompt learns task-specific representation for one task, while all the prompts can jointly contribute to the learning of the shared image token representations, and the interactions between different task prompts model the cross-task relationship. To decode dense predictions for multiple tasks with the learned spatial-channel task prompts from transformer, we accordingly design a dense task prompt decoding mechanism, which queries the shared image tokens using task prompts to obtain spatial- and channel-wise task-specific representations. Extensive experiments on two challenging multi-task dense scene understanding benchmarks (*i.e.* NYUD-V2 and PASCAL-Context) show the superiority of the proposed framework and TaskPrompter establishes significant state-of-the-art performances on multi-task dense predictions. Codes and models are publicly available at `https://github.com/prismformore/Multi-Task-Transformer`.

## 1 Introduction

Dense visual scene understanding is a fundamental research topic in computer vision that involves many dense prediction tasks, including semantic segmentation, depth estimation, surface normal estimation, boundary detection, etc. These distinct tasks share a fundamental understanding of the scene, which motivates researchers to design learning systems that model and predict multiple tasks in a unified framework, which is called "multi-task learning" (MTL). MTL mainly has two strengths: on one hand, learning a unified multi-task model for multiple tasks is typically more parameter-efficient than training several single-task models; on the other hand, different tasks can facilitate each other with a good design in MTL (Vandenhende et al., 2021).

With the powerful boost of deep learning, researchers have successfully designed highly promising multi-task learning models by exploiting the commonality and individuality of the tasks (Mani-

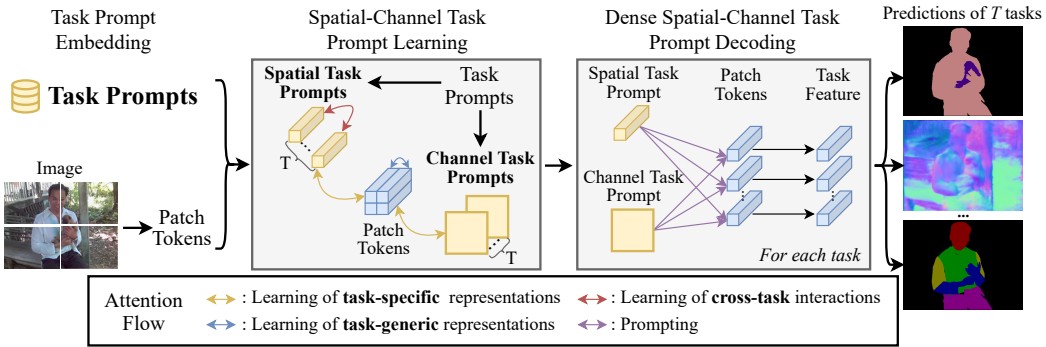

Figure 1: Illustration of our Spatial-Channel Multi-task Prompting framework (TaskPrompter). TaskPrompter unifies the learning of task-specific and task-generic representations as well as cross-task interactions in each layer throughout the whole transformer architecture, with the embedding of task prompts and and patch tokens. The task prompts are projected to spatial task prompts and channel task prompts to learn spatial- and channel-wise interactions, which are critical for dense predictions. The spatial and channel task prompts as well as patch tokens are further used in the proposed Dense Spatial-Channel Task Prompt Decoding module to prompt dense task-specific features and the final multi-task predictions.

nis et al., 2019; Xu et al., 2018; Kendall et al., 2018; Kokkinos, 2017). Traditionally, researchers manually design different types of modules in the multi-task network architecture to learn useful information for multi-task predictions on three aspects: task-generic representations, task-specific representations, and cross-task interactions. For instance, earlier works (Liu et al., 2019; Gao et al., 2019; Misra et al., 2016) design dedicated modules to learn task-specific representations and embed cross-task information interactions through hand-crafted structures deployed in the encoder, while several recent works (Ye & Xu, 2022; Li et al., 2022b; Vandenhende et al., 2020) choose to develop task-specific and cross-task modules in the decoder, and share encoder among different tasks. However, all of these methods decouple the learning of task-generic representations, task-specific representations, and cross-task interactions, into different network modules, which not only makes the architecture design more challenging as each module needs to be configured with a specific structure and capacity, but also suboptimal as learning effective communication among these three important perspectives of information is critical for multi-task dense prediction.

To tackle the above-mentioned issue, we believe a better MTL framework should be capable of learning task-generic and task-specific representations as well as their interactions jointly in each layer across the whole network architecture. In this paper, we achieve this goal by proposing a novel Spatial-Channel Multi-task Prompting framework, coined as TaskPrompter. The *core idea* of TaskPrompter is to design "spatial-channel task prompts" which are task-specific learnable tokens to learn spatial- and channel-wise task-specific information for each task. More specifically, the task prompts are embedded together with the task-generic patch tokens computed from the input image as input of a transformer with a specially designed Spatial-Channel Task Prompt Learning module. The task prompts and patch tokens interact with each other and refine themselves by means of attention mechanism in each transformer layer. In this way, TaskPrompter manages to learn task-generic and task-specific representation as well as cross-task interaction simultaneously and does not require the design of different types of network modules.

With the learned spatial-channel task prompts and image patch tokens, it is a non-trivial problem how to effectively decode multi-task dense features and predictions from them. To meet this challenge, we further propose a novel Dense Spatial-Channel Task Prompt Decoding method, which leverages both the spatial-wise and channel-wise affinities calculated between the task prompts and the patch tokens in attention modules to extract dense task features. The features are further refined by the cross-task affinity obtained from the self-attention weights among task prompts. The final multi-task dense predictions are produced based on the dense task features.

In summary, the contribution of this work consists of three parts:

- We propose a novel Spatial-Channel Multi-task Prompting framework (TaskPrompter) for multi-task dense scene understanding. Our method essentially combines the learning of task-generic and task-specific representations, as well as cross-task interactions in each layer across the whole network architecture by introducing task prompts in our transformer.

- A Spatial-Channel Task Prompt Learning module is designed. It can be flexibly deployed in each transformer layer for learning and refining task prompts and patch tokens along both spatial and channel dimensions.
- We further design a novel Dense Spatial-Channel Task Prompt Decoding method based on the learned task-specific task prompts and task-generic patch tokens to generate pixel-wise predictions for multiple tasks simultaneously.

Extensive experiments on two challenging multi-task dense prediction benchmarks (*i.e.* PASCAL-Context and NYUD-v2) clearly verify the effectiveness of the proposed method, which demonstrates superior performance compared with the previous state-of-the-art methods.

## 2 RELATED WORK

**Multi-task Dense Scene Understanding with Deep Learning** Several works have verified that scene understanding tasks can benefit from each other via multi-task learning (MTL) in deep learning era (Li et al., 2022a; Maninis et al., 2019; Kokkinos, 2017; Misra et al., 2016). MTL can also improve the computation efficiency of both training and inference compared with those with single-task models (Vandenhende et al., 2020). On the one hand, some researchers work on improving the optimization process of MTL, including loss design (Yang et al., 2023; Liu et al., 2022; Zamir et al., 2020; Kendall et al., 2018) and gradient manipulation (Chen et al., 2020; 2018). On the other hand, the architecture design of multi-task deep models is also widely explored (Vandenhende et al., 2021). We can divide most existing works on multi-task architecture design into two categories: encoder-based and decoder-based methods. Encoder-based methods (Gao et al., 2019; Liu et al., 2019; Misra et al., 2016) focus on designing cross-task interaction modules in the encoder, while the decoder-based methods embed these modules in the decoder stage. As the decoder-based methods can readily leverage the weights of backbone models pre-trained on large-scale image dataset (*e.g.* ImageNet), they generally yield stronger performance than encoder-based methods and have become mainstream pipelines for multi-task scene understanding (Bruggemann et al., 2021; Xu et al., 2018; Zhang et al., 2019; Zhou et al., 2020; Zhang et al., 2021). Recently, transformer models have been widely explored for multi-task learning (Ye & Xu, 2022; Bhattacharjee et al., 2022; Xu et al., 2022b; Liang et al., 2022; Xu et al., 2022a). Specifically, Ye & Xu (2022) propose a strong multi-task transformer "InvPT", which uses self-attention (Vaswani et al., 2017) to simultaneously learn spatial and cross-task relationships in a global context, establishing a strong multi-task performance. However, these existing methods decouple the learning of three important perspectives in multi-task learning, *i.e.* task-generic, task-specific representations, and cross-task interactions, into separate modules, which requires a careful design for each module, making the construction of the overall multi-task architecture challenging and suboptimal. To address this issue, this paper proposes a novel and effective multi-task prompting framework based on transformer to simultaneously learn these perspectives in each network layer in an end-to-end manner.

**Prompting** Prompting is a burgeoning paradigm designed for transformer. It is originally proposed to adapt pre-trained transformer models to downstream tasks by modifying the input of transformer layers without fine-tuning all the model parameters (Schick & Schütze, 2021; Gao et al., 2021; Lester et al., 2021; Brown et al., 2020; Shin et al., 2020). While some works try to explicitly design prompts (Shin et al., 2020; Petroni et al., 2019), more and more researchers turn to back-propagation for learning prompts (Lester et al., 2021; Li & Liang, 2021), which is termed as "Prompt Tuning". It typically embeds learnable prompts in the input token sequence of transformer and automatically tunes the prompts in training (Liu et al., 2021a; Hambardzumyan et al., 2021). Recently, the idea of prompting is introduced in Vision-language tasks (Zhou et al., 2022b; Ju et al., 2022; Yao et al., 2021; Ge et al., 2022; Zhou et al., 2022c; Radford et al., 2021), and further explored in visual tasks, such as image classification (Jia et al., 2022; Sandler et al., 2022; Bahng et al., 2022; Wang et al., 2022), open-vocabulary semantic segmentation (Rao et al., 2022; Zhou et al., 2022a), and object detection (Du et al., 2022; Feng et al., 2022). More related to our work, HyperPrompt (He et al., 2022) addresses multiple NLP tasks with prompts generated from task-specific hyper-networks. It is different from our proposal in three major ways: First, HyperPrompt needs to perform separate inferences for different tasks with different prompts; Second, because of the first limit, it has no cross-task interaction mechanism. Third, our special spatial-channel task prompting and decoding mechanisms are designed for dense predictions, which is totally different from NLP tasks. More-

over, our end-to-end multi-task prompting framework enables cross-task interactions throughout the whole network and generates pixel-wise predictions for all tasks with only one inference.

## 3  TASKPROMPTER: SPATIAL-CHANNEL MULTI-TASK PROMPTING

The proposed TaskPrompter framework (see Fig. 1) can be divided into three parts, *i.e.* Prompt Embedding, Spatial-Channel Task Prompt Learning, and Dense Spatial-Channel Task Prompt Decoding. We now introduce the details of these components one by one in this section.

### 3.1  PROMPT EMBEDDING

We adopt a classic transformer pipeline to embed an input image into a sequence of patch tokens (Dosovitskiy et al., 2021). It should be noted that our method is independent of the selection of different transformer architectures. The input image is first processed by a patch embedding layer, which is a convolutional layer with feature resolution downsampling. After the patch embedding layer, suppose the output feature map has a shape $(H, W, C)$ where $H$ and $W$ are the height and width, and $C$ is the number of channels of the feature map. The feature map is first reshaped into $N = H \times W$ patch tokens in a $C$-dimensional latent space and then added by positional encodings. To enable multi-task prompting, we propose to embed $T$ *learnable* task-specific tokens as **task prompts** in the same $C$-dimensional latent space as that of patch tokens, where $T$ is the number of tasks in multi-task learning. Each task prompt corresponds to a task. Next, the task prompts are concatenated with the patch tokens and form a token sequence matrix $\mathbf{Z}_0$:

$$\mathbf{Z}_0 = [\mathbf{p}^1; \mathbf{p}^2; \ldots; \mathbf{p}^T; \mathbf{x}^1; \mathbf{x}^2; \ldots; \mathbf{x}^N] \in \mathbb{R}^{(T+N) \times C}, N = H \times W, \tag{1}$$

where $[\cdot]$ indicates concatenation, $\{\mathbf{p}^i\}_{i=1}^T$ denotes the sequence of task prompts, and $\{\mathbf{x}^i\}_{i=1}^N$ denotes the sequence of patch tokens. In this way, the input image and task prompts are encoded and aggregated into a joint token sequence, which is refined through the proposed spatial-channel task prompt learning process.

### 3.2  SPATIAL-CHANNEL TASK PROMPT LEARNING

We design a spatial-channel task prompt learning module for concurrently learning and refining task prompts and patch tokens along both spatial and channel feature dimensions, as the spatial and channel information are both critical for various dense prediction tasks (Fu et al., 2019). The module details are depicted in Fig. 2. The goal of our design is to learn task-generic and task-specific visual representations, as well as cross-task interactions simultaneously, by taking full advantage of the structure of transformer with the embedded task prompts. Specifically, for each transformer layer, we adopt the basic layer structure of the classic ViT (Dosovitskiy et al., 2021) including MLPs, layer normalization layers (LayerNorm), and skip connections, while the core multi-head attention module is replaced by the proposed Spatial-Channel Task Prompt Learning module, in which, we design two types of task prompt learning paradigms: (i) Spatial Task Prompt Learning and (ii) Channel Task Prompt Learning.

Without loss of generality, we use the first transformer layer for illustration. After the first LayerNorm, we obtain a token sequence $\mathbf{Z} = \text{LayerNorm}(\mathbf{Z}_0) \in \mathbb{R}^{(T+N) \times C}$, which consists of task prompts $\mathbf{P} \in \mathbb{R}^{T \times C}$ and patch tokens $\mathbf{X} \in \mathbb{R}^{N \times C}$. Then, $\mathbf{P}$ is projected into spatial task prompts $\mathbf{P}^s$ and channel task prompts $\mathbf{P}^c$ where the superscript $s$ denotes 'spatial' and $c$ denotes 'channel', and the projection can be formulated as:

$$\mathbf{P}^s = f_{C \to C}(\mathbf{P}) \in \mathbb{R}^{T \times C}, \qquad \mathbf{P}^c = f_{C \to N}(\mathbf{P}) \in \mathbb{R}^{T \times N}, \tag{2}$$

where $f_{C \to C}$ denotes an identity mapping with the feature dimension unchanged; $f_{C \to N} : \mathbb{R}^{T \times C} \to \mathbb{R}^{T \times N}$ is a 2-layer MLP that projects the input task prompts $\mathbf{P}$ from $C$-dimensional latent space to $N$-dimensional latent space ($N = H \times W$). In this way, each channel task prompt aligns the feature dimension to the number of patch tokens for performing channel interactions. Then, $\mathbf{P}^s$ and $\mathbf{X}$ are fed into the proposed spatial task prompt learning module, and concurrently $\mathbf{P}^c$ and $\mathbf{X}$ are fed into the channel task prompt learning module.

**Spatial Task Prompt Learning** This module simultaneously learns spatial task prompts $\mathbf{P}^s$ and patch tokens $\mathbf{X}$, where each spatial task prompt interacts with patch tokens along the spatial dimension to model the spatial-wise relationships between task prompts and patch tokens. The spatial

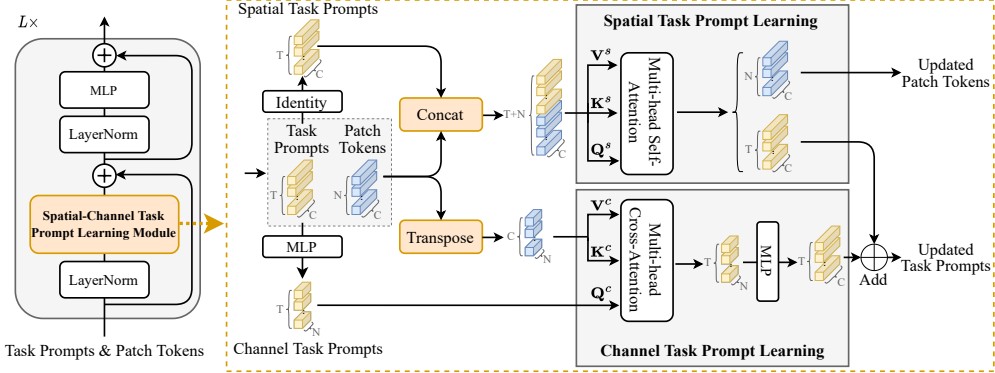

Figure 2: An illustration of the proposed Spatial-Channel Task Prompt Learning module in a transformer layer. This module learns the $T$ task prompts by interacting with patch tokens along both the spatial and channel dimensions. The task prompts are projected into $T$ spatial task prompts (each with $C$-dimensional) for Spatial Task Prompt Learning and into $T$ channel task prompts (each with $N$-dimensional) for Channel Task Prompt Learning.

task prompts $\mathbf{P}^s$ and patch tokens $\mathbf{X}$ are stacked as a token sequence $[\mathbf{P}^s; \mathbf{X}] \in \mathbb{R}^{(T+N) \times C}$, which is linearly projected by learnable parameters $\mathbf{W}_q^s, \mathbf{W}_k^s, \mathbf{W}_v^s \in \mathbb{R}^{C \times C}$, to respectively produce the query $\mathbf{Q}^s$, the key $\mathbf{K}^s$, and the value $\mathbf{V}^s$ as follows:

$$\mathbf{Q}^s = [\mathbf{P}^s; \mathbf{X}] \times \mathbf{W}_q^s, \quad \mathbf{K}^s = [\mathbf{P}^s; \mathbf{X}] \times \mathbf{W}_k^s, \quad \mathbf{V}^s = [\mathbf{P}^s; \mathbf{X}] \times \mathbf{W}_v^s. \quad (3)$$

As a standard procedure, before we compute the multi-head self-attention (MSA), we need to partition $\mathbf{Q}^s, \mathbf{K}^s, \mathbf{V}^s$ evenly along the last dimension into different groups as input of different heads. Suppose that we utilize $N_{head}^s$ heads in MSA, after head partition we have $\mathbf{Q}^s, \mathbf{K}^s, \mathbf{V}^s \in \mathbb{R}^{N_{head}^s \times (T+N) \times \frac{C}{N_{head}^s}}$ and spatial self-attention map is computed by $\mathbf{A}^s = \mathbf{Q}^s \times \mathbf{K}^{s\top}$, where the symbol '$\top$' denotes a transposing operation that transposes the last two dimensions of a tensor. $\mathbf{A}^s$ is then scaled and normalized by a softmax function, and multiplied by $\mathbf{V}^s$ to obtain a new token sequence. The token sequence merges different heads and is projected by a linear layer as in a standard MSA, which provides us an output token sequence, consisting of updated spatial task prompts $\mathbf{P}^{s\prime}$ and patch tokens $\mathbf{X}'$.

**Channel Task Prompt Learning** This module is proposed to model the channel-wise relationships between the channel task prompts $\mathbf{P}^c$ and the patch tokens $\mathbf{X}$ along the channel dimension. We perform cross-attention to model the channel-wise relationships. The channel task prompts $\mathbf{P}^c$ are projected by a learnable parameter $\mathbf{W}_q^c \in \mathbb{R}^{N \times N}$ to produce the query $\mathbf{Q}^c \in \mathbb{R}^{T \times N}$, and the patch tokens $\mathbf{X}$ are separately projected by two learnable parameter matrices $\mathbf{W}_k^c \in \mathbb{R}^{C \times C}$ and $\mathbf{W}_v^c \in \mathbb{R}^{C \times C}$, and transposed to produce the key $\mathbf{K}^c$ and the value $\mathbf{V}^c$:

$$\mathbf{Q}^c = \mathbf{P}^c \times \mathbf{W}_q^c, \quad \mathbf{K}^c = (\mathbf{X} \times \mathbf{W}_k^c)^\top, \quad \mathbf{V}^c = (\mathbf{X} \times \mathbf{W}_v^c)^\top. \quad (4)$$

Before computing the multi-head cross-attention, we need to partition $\mathbf{Q}^c, \mathbf{K}^c, \mathbf{V}^c$ along the last dimension into different groups as the input of different heads in multi-head channel-wise cross attention. Suppose the number of heads used in Channel Task Prompt Learning is $N_{head}^c$, the most straightforward way is evenly partitioning the matrices along their last dimension into $N_{head}^c$ groups. However, as $\mathbf{K}^c$ and $\mathbf{V}^c$ are computed from patch tokens $\mathbf{X}$, the last dimension of them contains the spatial relationship of pixels, while a standard partition method disrupts this spatial relationship which is critical for learning features for dense predictions. A more reasonable strategy is reorganizing $\mathbf{Q}^c, \mathbf{K}^c, \mathbf{V}^c$ based on the spatial adjacency. Specifically, we first reshape these matrices to a spatial shape $\mathbb{R}^{C \times H \times W}$ as $H \times W = N$, and then partition the spatial planes formed by the last 2 dimensions evenly into $N_{head}^c$ local windows, as shown in Fig. 6. Notably, $N_{head}^c$ needs to be properly set so that the number of windows along the height dimension $N_{win}^h$ and the width dimension $N_{win}^w$ satisfies $N_{win}^h \times N_{win}^w = N_{head}^c$. This process is the proposed "window partition" to maintain the spatial relationship for multi-head channel cross-attention calculation. After the window partition, we have $\mathbf{Q}^c \in \mathbb{R}^{N_{head}^c \times T \times N/N_{head}^c}, \mathbf{K}^c, \mathbf{V}^c \in \mathbb{R}^{N_{head}^c \times C \times N/N_{head}^c}$. Then, channel attention maps are calculated by $\mathbf{A}^c = \mathbf{Q}^c \times \mathbf{K}^{c\top}$. $\mathbf{A}^c$ is scaled and normalized by a softmax function and multiplied by $\mathbf{V}^c$ to obtain the updated channel task prompts $\mathbf{P}^{c\prime}$ after being processed by a linear layer. To update the overall task prompts $\mathbf{P}$ with information learned from both the spatial and channel task prompts, as shown in Fig. 2, $\mathbf{P}^{c\prime}$ is projected back to the $C$-dimensional latent space with a

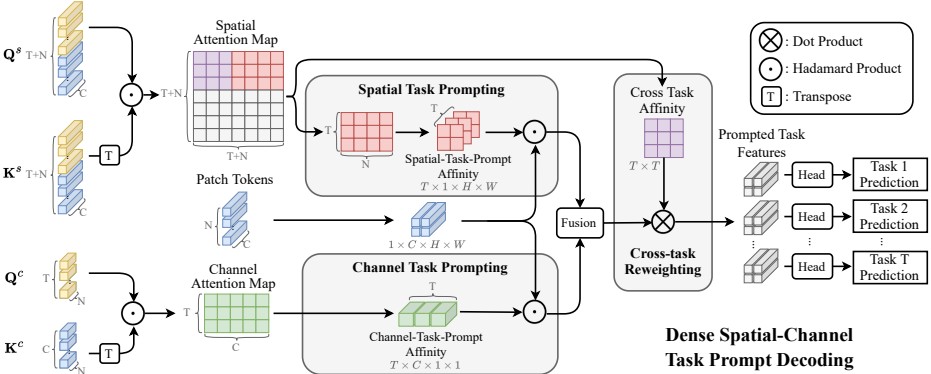

Figure 3: A diagram illustration of Dense Spatial-Channel Task Prompt Decoding. The spatial attention map and channel attention map are calculated from the query and key tensors in Spatial Task Prompt Learning and Channel Task Prompt Learning, respectively. They are used to guide the decoding of task-specific features from patch tokens along spatial and channel dimensions.

2-layer MLP $f_{N \to C} : \mathbb{R}^{T \times N \to T \times C}$, and then added by $\mathbf{P}^{s\prime}$ to obtain a combined task prompts $\mathbf{P}'$ as an update of $\mathbf{P}$ as follows:

$$\mathbf{P}' = \mathbf{P}^{s\prime} + f_{N \to C}(\mathbf{P}^{c\prime}). \tag{5}$$

$\mathbf{P}'$ and $\mathbf{X}'$ are refined by the typical LayerNorm and MLP, and then stacked as a new token sequence $\mathbf{Z}'$, which is fed into the next transformer layer following the same procedure for further learning.

### 3.3 DENSE SPATIAL-CHANNEL TASK PROMPT DECODING

To decode multiple dense predictions for distinct tasks from the task-specific task prompts and task-generic patch tokens, we need to design an effective decoding method for TaskPrompter. Since the task prompts including the spatial and channel task prompts are task-discriminative, then the affinities calculated between the spatial/channel task prompts and the shared patch tokens are also distinct. The different task prompts localize different spatial regions or channels on the patch tokens. This can also be confirmed from our visualization of the learned spatial and channel affinities, as shown in Fig. 4. Based on the learned spatial and channel task prompts, we propose a Dense Spatial-Channel Task Prompt Decoding method, which consists of Spatial Task Prompting and Channel Task Prompting strategies, to respectively compute spatial-wise and channel-wise task-specific features, for the final multi-task predictions, as shown in Fig. 3.

**Spatial Task Prompting** Each spatial task prompt corresponds to a spatial affinity map by computing the affinity between the spatial task prompt and all the patch tokens. We denote this spatial affinity map as Spatial-Task-Prompt Affinity, as shown in Fig. 3. It can be extracted directly from the learned spatial attention maps, *i.e.* $\mathbf{A}^s \in \mathbb{R}^{N^s_{head} \times (T+N) \times (T+N)}$ with $N = H \times W$ as the number of patch tokens and $T$ as the number of tasks, in the task-prompt learning stage. For the spatial task prompt of task $t$, we can extract an attention tensor in space $\mathbb{R}^{N^s_{head} \times N}$ from $\mathbf{A}^s$, and then we reshape it to be a new tensor in space $\mathbb{R}^{N^s_{head} \times 1 \times H \times W}$, which is the Spatial-Task-Prompt Affinity and we denote it as $\mathbf{A}^{p \to s}_t$. On the other hand, given an updated token sequence $\mathbf{X}' \in \mathbb{R}^{N \times C}$ produced from a transformer layer, we transpose and reshape $\mathbf{X}'$ into space $\mathbb{R}^{N^s_{head} \times C/N^s_{head} \times H \times W}$, and denote it as $\mathbf{X}'^s$. The spatial-wise task-specific features $\mathbf{F}^s_t$ for task $t$ can be decoded by:

$$\mathbf{F}^s_t = f_{sr}(\mathbf{A}^{p \to s}_t \odot \mathbf{X}'^s), \tag{6}$$

where the symbol $\odot$ indicates the operation of a Hadamard product and $f_{sr}(\cdot)$ represents the operation of reshaping the tensor into space $\mathbb{R}^{C \times H \times W}$.

**Channel Task Prompting** Each channel task prompt corresponds to a channel affinity vector, which can be computed by measuring the affinity between the channel task prompt and all the channels of the patch tokens. We name it Channel-Task-Prompt Affinity, denoted as $\mathbf{A}^{p \to c}_t$ with $t$ indicating the task $t$, which can be used to decode the channel-wise task-specific representation for task $t$. We obtain the channel-task-prompt affinity $\mathbf{A}^{p \to c}_t$ of task $t$ from $\mathbf{A}^c \in \mathbb{R}^{N^c_{head} \times T \times C}$ by slicing one along the second dimension and reshaping it into space $\mathbb{R}^{C \times N^h_{win} \times 1 \times N^w_{win} \times 1}$. We also reshape $\mathbf{X}'$ as $\mathbf{X}'^c \in \mathbb{R}^{C \times N^h_{win} \times h_{win} \times N^w_{win} \times w_{win}}$, where $h_{win} = H/N^h_{win}$ and $w_{win} = W/N^w_{win}$. Then, we can compute the channel-wise task-specific features $\mathbf{F}^c_t$ for task $t$ as follows:

$$\mathbf{F}^c_t = f_{cr}(\mathbf{A}^{p \to c}_t \odot \mathbf{X}'^c), \tag{7}$$

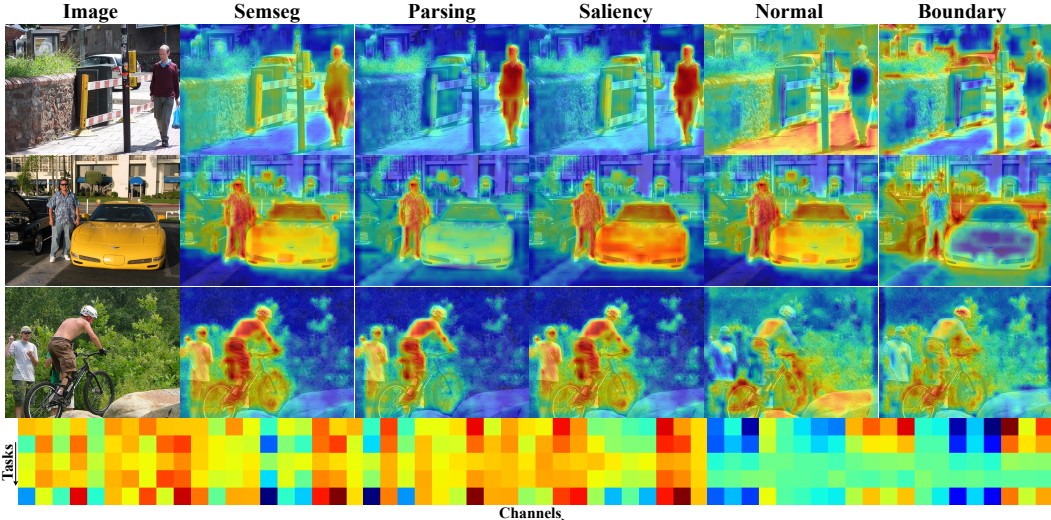

Figure 4: Visualization examples of the spatial-task-prompt affinity (the first three rows) and the channel-task-prompt affinity (the last row). It can be observed that different spatial and channel task prompts can both attend to distinct spatial or channel locations of the patch tokens, which indicates that the task prompts can effectively learn task-specific representations from the interaction with the image patch tokens.

where $f_{cr}(\cdot)$ denotes the operation of reshaping the tensor into the space $\mathbb{R}^{C \times H \times W}$.

**Spatial-Channel Fusion** To fuse the task-specific features $\mathbf{F}_t^s$ and $\mathbf{F}_t^c$ from the spatial and channel task prompting, we concatenate them along the channel dimension and reduce the channel number by half via using a $3 \times 3$ convolution ($\text{CONV}_{3\times3}$) with batch normalization (BN) and GELU to obtain a fused task-specific feature $\mathbf{F_t}$ for task $t$ as follows:

$$\mathbf{F}_t = \text{GELU} \circ \text{BN} \circ \text{CONV}_{3\times3}([\mathbf{F}_t^s; \mathbf{F}_t^c]). \tag{8}$$

Then, we stack the prompted task-specific features of all the $T$ tasks along the first dimension and obtain an overall task feature map $\mathbf{F} \in \mathbb{R}^{T \times C \times H \times W}$.

**Cross-task Reweighting** The spatial and channel task prompting decode the relationship between task prompts and patch tokens. The cross-task relationship is not involved, while the cross-task relationship is also modeled in the task-prompt learning stage in the encoder. To encourage cross-task information exchange in our decoding stage, we further put forward Cross-task Reweighting as also shown in Fig. 3. First, we extract the affinity tensor among $T$ task prompts from the attention map $\mathbf{A}^s$. The affinity tensor is in space $\mathbb{R}^{N_{head}^s \times T \times T}$. Then, we project the first dimension to 1 using a 2-layer MLP, and obtain the Cross Task Affinity $\mathbf{A}^{p \to p}$. The prompted task features $\mathbf{F} \in \mathbb{R}^{T \times C \times H \times W}$ are updated by $\mathbf{F} \leftarrow \mathbf{A}^{p \to p} \times \mathbf{F}$. The prompted task features $\mathbf{F}$ contain features for all the $T$ tasks. They are split and separately fed into $T$ task-specific prediction heads for dense predictions. Each prediction head is composed of a simple $3 \times 3$ convolutional block with BN and GELU, and a linear projection layer.

**Hierarchical Prompting** As discussed by previous multi-task transformer for dense scene understanding (Ye & Xu, 2022), different levels of transformer features help improve the multi-task performance. Therefore, we deploy our prompt decoding method to multiple levels of the transformer, and name it "Hierarchical Prompting (HP)". Specifically, we conduct Dense Spatial-Channel Task Prompt Decoding at multiple levels of transformer for prompting the task features, instead of only the last layer. The multi-level task features are later fused as one by addition to obtain the final prompted task features, which are fed into the prediction heads as described above.

# 4 EXPERIMENTS

## 4.1 EXPERIMENTAL SETUP

**Datasets** We evaluate the proposed TaskPrompter mainly on two mostly used multi-task dense visual scene understanding datasets, *i.e.* **NYUD-v2** (Silberman et al., 2012) and **PASCAL-Context** (Chen et al., 2014). Details of the datasets are presented in Appendix A.3.

Table 1: Effectiveness of different components of TaskPrompter. The performance gains compared against the baseline are shown in brackets. '↓' means lower better and '↑' means higher better.

| Model | Semseg mIoU ↑ | Parsing mIoU ↑ | Saliency maxF ↑ | Normal mErr ↓ | Boundary odsF ↑ | MTL Gain $\Delta_m$ ↑ |
|---|---|---|---|---|---|---|
| STL Model | 78.42 | 68.36 | 85.34 | 13.87 | 73.90 | - |
| **TaskPrompter** Baseline | 75.04 | 64.82 | 84.59 | 14.17 | 68.00 | -4.11 |
| +SPrompt | 75.95 (↑ **0.91**) | 65.60 (↑ **0.78**) | 84.91 (↑ **0.31**) | 13.97 (↓ **0.20**) | 70.50 (↑ **2.50**) | -2.59 (↑ **1.52**) |
| +SPrompt + CPrompt | 76.46 (↑ **1.41**) | 66.25 (↑ **1.43**) | 85.00 (↑ **0.41**) | 13.97 (↓ **0.20**) | 71.10 (↑ **3.10**) | -2.11 (↑ **2.00**) |
| +SPrompt + CPrompt + RW | 76.83 (↑ **1.79**) | 66.31 (↑ **1.49**) | 85.00 (↑ **0.41**) | 13.94 (↓ **0.23**) | 71.30 (↑ **3.30**) | -1.90 (↑ **2.21**) |
| +SPrompt + CPrompt + RW + HP | 79.00 (↑ **3.96**) | 67.00 (↑ **2.18**) | 85.05 (↑ **0.46**) | 13.47 (↓ **0.70**) | 73.50 (↑ **5.50**) | 0.15 (↑ **4.26**) |

Table 2: Comparison between TaskPrompter-Base and TaskPrompter-Large.

| Model | PASCAL-Context | | | | | NYUD-v2 | | | |
|---|---|---|---|---|---|---|---|---|---|
| | Semseg mIoU ↑ | Parsing mIoU ↑ | Saliency maxF ↑ | Normal mErr ↓ | Boundary odsF ↑ | Semseg mIoU ↑ | Depth RMSE ↓ | Normal mErr ↓ | Boundary odsF ↑ |
| TaskPrompter-Base | 79.00 | 67.00 | 85.05 | 13.47 | 73.50 | 50.40 | 0.5402 | 18.91 | 77.60 |
| TaskPrompter-Large | 80.89 | 68.89 | 84.83 | 13.72 | 73.50 | 55.30 | 0.5152 | 18.47 | 78.20 |

**Evaluation Metrics** Semantic segmentation (Semseg) and human parsing (Parsing) use the mean Intersection over Union (mIoU) metric for evaluation. Monocular depth estimation (Depth) uses Root Mean Square Error (RMSE). Surface normal estimation (Normal) uses mean error (mErr) of predicted angles. Saliency detection (Saliency) adopts maximal F-measure (maxF) for evaluation. Object boundary detection (Boundary) is evaluated with the optimal-dataset-scale F-measure (odsF). Multi-task gain (MTL Gain) is evaluated by metric $\Delta_m$ introduced in (Maninis et al., 2019).

**Models Declaration** To verify the model effectiveness, we consider the following model variants in Table 1: **(i)** "TaskPrompter Baseline" is a very strong baseline with comparable performance to the-state-of-the-arts for multi-task dense predictions. It is built upon ViT-Base with 12 transformer layers pre-trained on ImageNet-22K, and uses task-specific Conv(3x3)-BN-ReLU-Conv(1x1) blocks as heads for final multi-task predictions. **(ii)** "STL Model" is a counterpart of (i) under single-task setting with the same network structure. It is used to train a set of single-task models, and each model is only trained for one task at a time. **(iii)** "TaskPrompter +SPrompt +CPrompt +RW +HP" denotes a full version of TaskPrompter using all the proposed modules. It is built on (i) with the same transformer architecture and parameter initialization. It uses 16 heads for Channel Task Prompt Learning. In Hierarchical Prompting (HP), it selects every third layer, in total 4 layers, to conduct Dense Spatial-Channel Task Prompt Decoding. **(iv)** "TaskPrompter +SPrompt +CPrompt +RW" removes Hierarchical Prompting from (iii). **(v)** "TaskPrompter +SPrompt +CPrompt" removes cross-task reweighting from (iv). **(vi)** "TaskPrompter +SPrompt" further disables Channel Task Prompt Learning from (v).

## 4.2 EXPERIMENTAL RESULTS

**Effectiveness of Spatial and Channel Task Prompt Learning and Decoding** We evaluate the proposed methods on the PASCAL-Context dataset and report the results in Table 1. Using Spatial Task Prompt Learning can already bring a clear performance improvement on all the tasks, particularly, with a boost of 0.91, 0.78, and 2.50 points for Semseg, Parsing, and Boundary respectively, compared against the baseline. By adding the Channel Task Prompt Learning, the performance gains are further increased to 1.41, 1.48, and 3.10 points for the three tasks, respectively. These experimental results clearly demonstrate the effectiveness of the core design of TaskPrompter.

**Effectiveness of Cross-Task Reweighting and Hierarchical Prompting** Furthermore, as shown in Table 1, cross-task reweighting can improve the performance of most of the tasks, with a multi-task gain (i.e. $\Delta_m$) of 0.21 points. Hierarchical Prompting helps increase the performance largely on all the tasks, by deploying task-prompt decoding on multiple levels of the transformer. Hierarchical Prompting is designed to facilitate the decoding of task-specific features with the learned spatial-channel task prompts at each layer of the transformer encoder. With the embedding and learning of the global task prompts from the beginning of the transformer encoder, our model can naturally perform spatial-channel hierarchical prompting at different layers for decoding task-specific features, which is very beneficial for producing more effective multi-task representations.

**Scaling TaskPrompter** We follow ViT (Dosovitskiy et al., 2021) and build the proposed Spatial-Channel Multi-task Prompting framework on a transformer with 24 layers, denoted as TaskPrompter-Large. We also denote the one with 12 layers as TaskPrompter-Base, and compare their performances on both PASCAL-Context and NYUD-v2 datasets. The results are reported in Table 2. We can observe that models with bigger capacity generally bring better performance for

Figure 5: Influence of the number of heads (windows) in Channel Task Prompt Learning.

Table 3: Comparison with state-of-the-arts on NYUD-v2 (*left*) and PASCAL-Context (*right*). Our TaskPrompter clearly outperforms the previous state-of-the-arts.

| Model | Semseg mIoU ↑ | Depth RMSE ↓ | Normal mErr ↓ | Boundary odsF ↑ |
|---|---|---|---|---|
| Cross-Stitch (Misra et al., 2016) | 36.34 | 0.6290 | 20.88 | 76.38 |
| PAP (Zhang et al., 2019) | 36.72 | 0.6178 | 20.82 | 76.42 |
| PSD (Zhou et al., 2020) | 36.69 | 0.6246 | 20.87 | 76.42 |
| PAD-Net (Xu et al., 2018) | 36.61 | 0.6270 | 20.85 | 76.38 |
| MTI-Net (Vandenhende et al., 2020) | 45.97 | 0.5365 | 20.27 | 77.86 |
| ATRC (Bruggemann et al., 2021) | 46.33 | 0.5363 | 20.18 | 77.94 |
| InvPT (Ye & Xu, 2022) | 53.56 | 0.5183 | 19.04 | 78.10 |
| **TaskPrompter (ours)** | **55.30** | **0.5152** | **18.47** | **78.20** |

| Model | Semseg mIoU ↑ | Parsing mIoU ↑ | Saliency maxF ↑ | Normal mErr ↓ | Boundary odsF ↑ |
|---|---|---|---|---|---|
| ASTMT (Maninis et al., 2019) | 68.00 | 61.10 | 65.70 | 14.70 | 72.40 |
| PAD-Net (Xu et al., 2018) | 53.60 | 59.60 | 65.80 | 15.30 | 72.50 |
| MTI-Net (Vandenhende et al., 2020) | 61.70 | 60.18 | 84.78 | 14.23 | 70.80 |
| ATRC (Bruggemann et al., 2021) | 62.69 | 59.42 | 84.70 | 14.20 | 70.96 |
| ATRC-ASPP (Bruggemann et al., 2021) | 63.60 | 60.23 | 83.91 | 14.30 | 70.86 |
| ATRC-BMTAS (Bruggemann et al., 2021) | 67.67 | 62.93 | 82.29 | 14.24 | 72.42 |
| InvPT (Ye & Xu, 2022) | 79.03 | 67.61 | 84.81 | 14.15 | 73.00 |
| **TaskPrompter (ours)** | **80.89** | **68.89** | **84.83** | **13.72** | **73.50** |

most of the tasks, but as discussed by Ye & Xu (2022); Kendall et al. (2018), the performances of some tasks may be worse because of the multi-task competition issue.

**Qualitative Visualization of Spatial-Channel Task Prompt Affinity** To investigate whether the task prompts learn task-specific affinity on patch tokens, we visualize the affinity values between task prompts and patch tokens in the Dense Spatial-Channel Task Prompt Decoding module as shown in Fig. 4 and Fig. 8. We can clearly observe that the activated spatial-task-prompt affinity values are highly related to the particularity of each task, which indicates that the spatial task prompts can effectively encode task-specific representations, and attend to different semantic regions of patch tokens that are more beneficial for the prediction of a specific task when performing the decoding. On the other hand, we calculate the average of channel-task-prompt affinity maps from all the test images of PASCAL-Context dataset and randomly select 60 channels for visualization. We can observe that the different channel task prompts have distinct attention responses to different channels, which verifies that channel-task-prompt affinity encodes task-specific relationships between channel task prompts and patch tokens along the channel dimension.

**Study of Number of Heads in Channel Task Prompt Learning** As introduced in Section 3.2, we partition query and key tensors in Channel Task Prompt Learning into different groups as input of different heads in cross-attention. We report the performance comparison of using different numbers of heads in Fig. 5. We observe that using more heads brings better performance for most tasks.

**Comparison with Previous SOTA on NYUD-v2 and PASCAL-Context** Table 3 reports a comparison of the proposed TaskPrompter against previous state-of-the-art methods, including InvPT (Ye & Xu, 2022), ATRC (Bruggemann et al., 2021), MTI-Net (Vandenhende et al., 2020) and PAD-Net (Xu et al., 2018), on both NYUD-v2 and PASCAL-Context datasets. Notably, the previous best method (*i.e.* InvPT) and our TaskPrompter are built upon the transformer architecture with the same backbone. Our TaskPrompter establishes new state-of-the-art performances on all 9 metrics on these two datasets. On NYUD-v2, the performance of Semseg is clearly boosted from the previous best, *i.e.* 53.56 to 55.30 (**+1.74**). On PASCAL-Context dataset, Semseg is improved from the previous best 79.03 to 80.89 (**+1.86**) and Parsing is improved from 67.61 to 68.89 (**+1.28**).

## 5 CONCLUSION

We have presented a Spatial-Channel Multi-task Prompting (TaskPrompter) framework for simultaneously learning task-generic and task-specific representations as well as cross-task interaction in each layer throughout the whole transformer architecture. We first propose to learn task prompts to encode task-specific information, and design a dedicated module to learn the relationship between task prompts and patch tokens along both spatial and channel dimensions. Furthermore, we propose a novel spatial-channel task prompt decoding method to generate dense task-specific features for prediction. The effectiveness of our method is validated by both quantitative and qualitative experiments, showing superior performances on different task sets. The performances of our method clearly surpass the existing multi-task dense scene understanding models.

## ACKNOWLEDGEMENTS

This research is supported in part by HKUST-SAIL joint research funding, the Early Career Scheme of the Research Grants Council (RGC) of the Hong Kong SAR under grant No. 26202321 and HKUST Startup Fund No. R9253.

## ETHICS STATEMENT

The proposed method focuses on the improvement of multi-task deep learning algorithms without introducing new tasks or datasets. Thus our work doesn't raise any new ethical issues.

## REPRODUCIBILITY STATEMENT

The experiments conducted in this paper are based on widely used public datasets. We present the proposed methods in detail and include the implementation particulars in Section 4.1 and Section A.3. The code will be made publicly available to help further study.

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

## A    APPENDIX

### A.1    PRELIMINARIES: MULTI-HEAD SELF-ATTENTION AND MULTI-HEAD CROSS-ATTENTION

Multi-head self-attention and multi-head cross-attention are both widely used variants of the attention mechanism proposed by Vaswani et al. (2017). As the input of the attention module, the token sequences are projected by individual learnable weights to query $\mathbf{Q}$, key $\mathbf{K}$, and value $\mathbf{V}$ matrices. Essentially, attention mechanism is a weighted addition operation on the value guided by the affinity between query and key values. The output of attention module is calculated by:

$$\text{Attention}(\mathbf{Q}, \mathbf{X}) = \text{softmax}\left(\frac{\mathbf{Q}\mathbf{K}^\top}{\sqrt{C}}\right) \times \mathbf{V}, \tag{9}$$

where $C$ is the embedding dimension. If the input token sequences are projected into $N_{head}$ set of query, key, and value tensors and compute attention separately, it is called "multi-head attention". The difference between multi-head self-attention (MHSA) and multi-head attention cross-attention (MHCA) is that the query, key, and value input of MHSA are projected from the same token sequence, while the query and key matrices of MHCA are projected from different token sequences.

### A.2    WINDOW PARTITION IN CHANNEL TASK PROMPT LEARNING

We show a visual illustration of the window partition technique used in Channel Task Prompt Learning in Fig. 6.

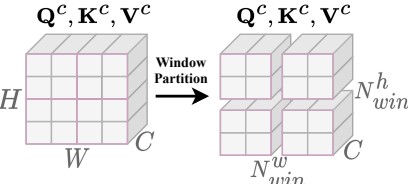

Figure 6: We partition the spatial planes formed by the last 2 dimensions of $\mathbf{K}^c$ and $\mathbf{V}^c$ evenly into local windows to maintain the spatial relationship when fed into different attention heads for cross-attention calculation in the Channel Task Prompt Learning.

### A.3    IMPLEMENTATION DETAILS

**Datasets** We evaluate the proposed TaskPrompter mainly on two mostly used multi-task dense visual scene understanding datasets, *i.e.* **NYUD-v2** (Silberman et al., 2012) and **PASCAL-Context** (Chen et al., 2014). Specifically, PASCAL-Context provides 4,998 images in the training set and 5,105 images in the testing set. This dataset offers dense labels for multiple tasks including semantic segmentation, human parsing, and object boundary detection. Additionally, Maninis et al. (2019) provide pseudo ground truth labels for surface normals estimation and saliency detection. On the other hand, NYUD-v2 totally provides 1,449 images, in which 795 are used for training and the rest 654 for testing. It includes dense labels for tasks including semantic segmentation, monocular depth estimation, surface normal estimation, and object boundary detection. In our experimental setup, we include all the tasks in these datasets for a comprehensive performance study.

**Model Training** The models for different experiments are trained for 40,000 iterations on all datasets, with a batch size of 4 if not otherwise specified. Adam optimizer is adopted with a learning rate of $2 \times 10^{-5}$, and a weight decay rate of $1 \times 10^{-6}$. A polynomial learning rate scheduler is used during optimization. For the continuous regression tasks (*i.e.* Depth and Normal) we use $\mathcal{L}1$ Losses. For the discrete classification tasks (*i.e.* Semseg, Parsing, Saliency, and Boundary) we use cross-entropy losses for them. The learnable task prompts are randomly initialized with normal distribution (mean=1, std=1).

**Data Processing.** For a fair comparison with Invpt (Ye & Xu, 2022), we follow its data processing pipeline. On PASCAL-Context, we pad the image to the size of $512 \times 512$, while on NYUD-v2, we randomly crop the input image to the size of $448 \times 576$. We use the same data augmentation including random color jittering, random cropping, random scaling, and random horizontal flipping.

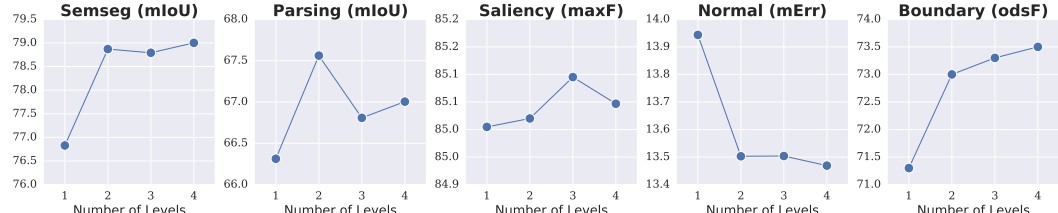

Figure 7: Influence of using different numbers of levels in Hierarchical Prompting (HP). The levels are chosen evenly based on network depth. "1" means not using HP. Using only two levels in HP can already bring large performance gain.

## A.4 Ablation Study of Using Different Numbers of Levels in Hierarchical Prompting

We investigate how the number of levels used by Hierarchical Prompting (HP) influences the overall performance of TaskPrompter. The experimental results with the varying number of levels are presented in Fig. 7. It can be observed that using only two levels in HP can already largely boost the performances on all the five tasks, demonstrating the effectiveness of the hierarchical prompting scheme. Further increasing the number of levels, it helps further improve some tasks (*e.g.*, Semseg, Normal, and Boundary), while for some others (*e.g.*, Parsing and Saliency), it shows saturated performances with small variances.

## A.5 Qualitative Study

**More Qualitative Visualization of Learned Spatial-Task-Prompt Affinity Maps** We provide more qualitative results by visualizing the learned Spatial-Task-Prompt Affinity maps on the PASCAL-Context dataset, as shown in Fig. 8. We can observe that the task prompts has very diverse affinities to different positions of the image patch tokens, demonstrating task-discriminative characteristics of the learned task prompts, and also verifying our motivation that the learned task prompts can directly query the task-generic image patch tokens to decode task-specific representations.

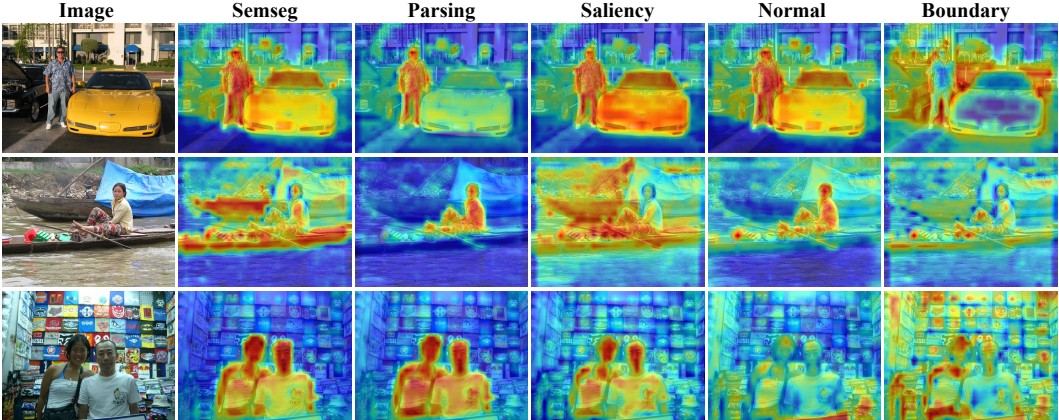

Figure 8: More visualization results of the spatial-task-prompt affinity maps on the PASCAL-Context dataset. We can observe that the task prompts attend to different areas of the images based on the characteristics of the tasks.

**Qualitative Comparison with the Best Performing Transformer-based Method.** We qualitatively compare the produced predictions of the proposed TaskPrompter with the best performing transformer-based method in the literature, *i.e.* InvPT (Ye & Xu, 2022). As shown in Fig. 9 and Fig. 10, our TaskPrompter model generates dense predictions of multiple distinct tasks with clearly better details than the InvPT as marked in circles.

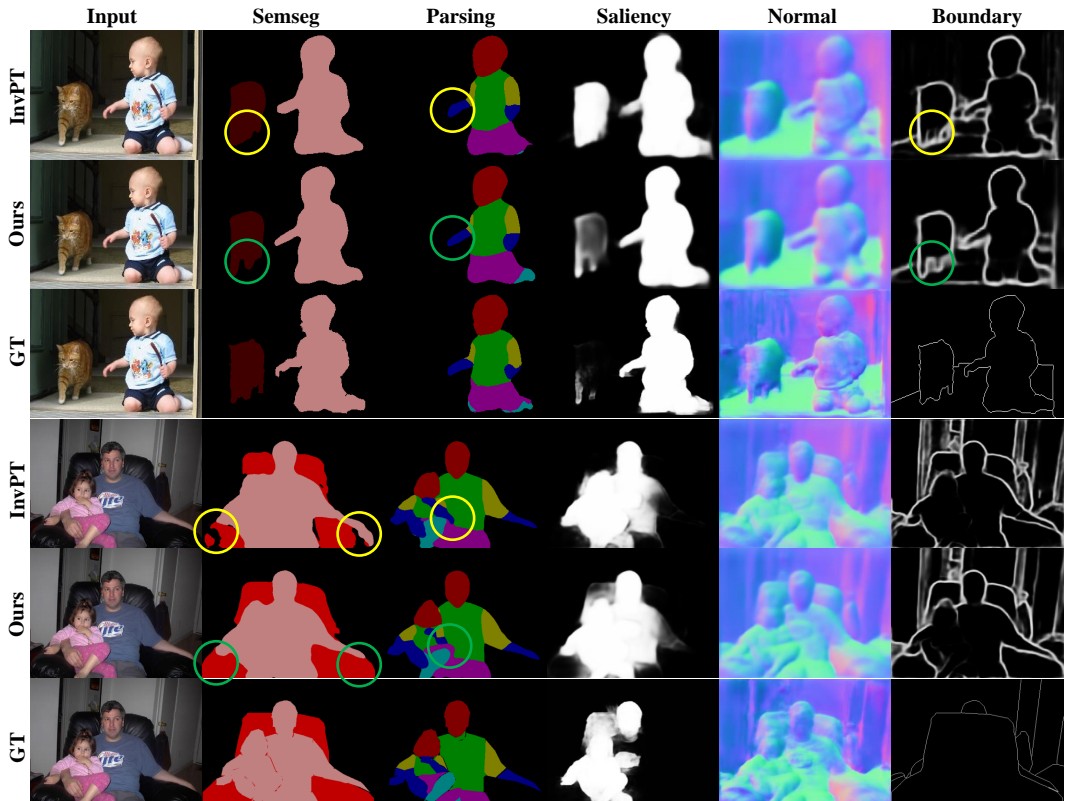

Figure 9: Qualitative comparison with the best performing transformer-based method (*i.e.* InvPT) on PASCAL-Context. TaskPrompter generates predictions with better details.

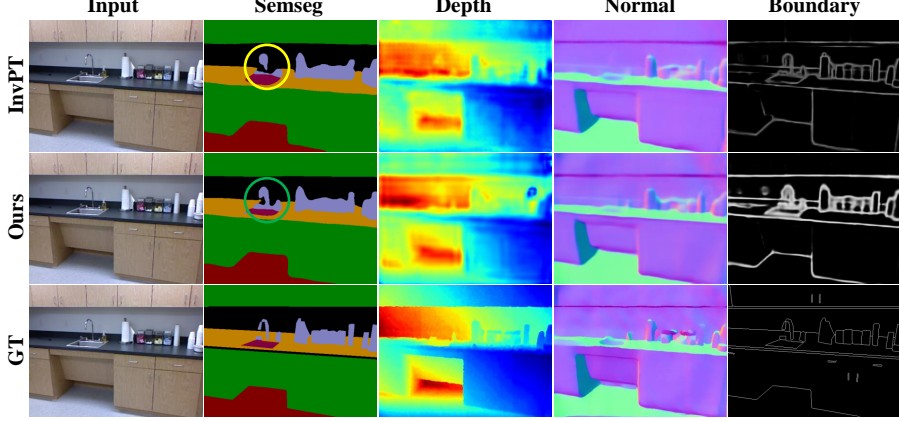

Figure 10: Qualitative comparison with the best performing transformer-based method (*i.e.* InvPT) on NYUD-v2. TaskPrompter generates better multi-task predictions.

## A.6    JOINT 2D-3D MULTI-TASK SCENE UNDERSTANDING ON CITYSCAPES-3D

To further examine the proposed TaskPrompter, we adapt it to tackle a joint 2D-3D multi-task scene understanding problem involving three challenging tasks, *i.e.* 3D object detection (3Ddet), semantic segmentation (Semseg), and monocular depth estimation (Depth), on Cityscapes-3D dataset (Gählert et al., 2020).

**Dataset** Cityscapes-3D (Gählert et al., 2020) is an extension of Cityscapes dataset (Cordts et al., 2016) via providing additional 3D bounding box annotations. Cityscapes-3D dataset consists of 2,975 training images and 500 validation images with fine annotations. The image resolution is

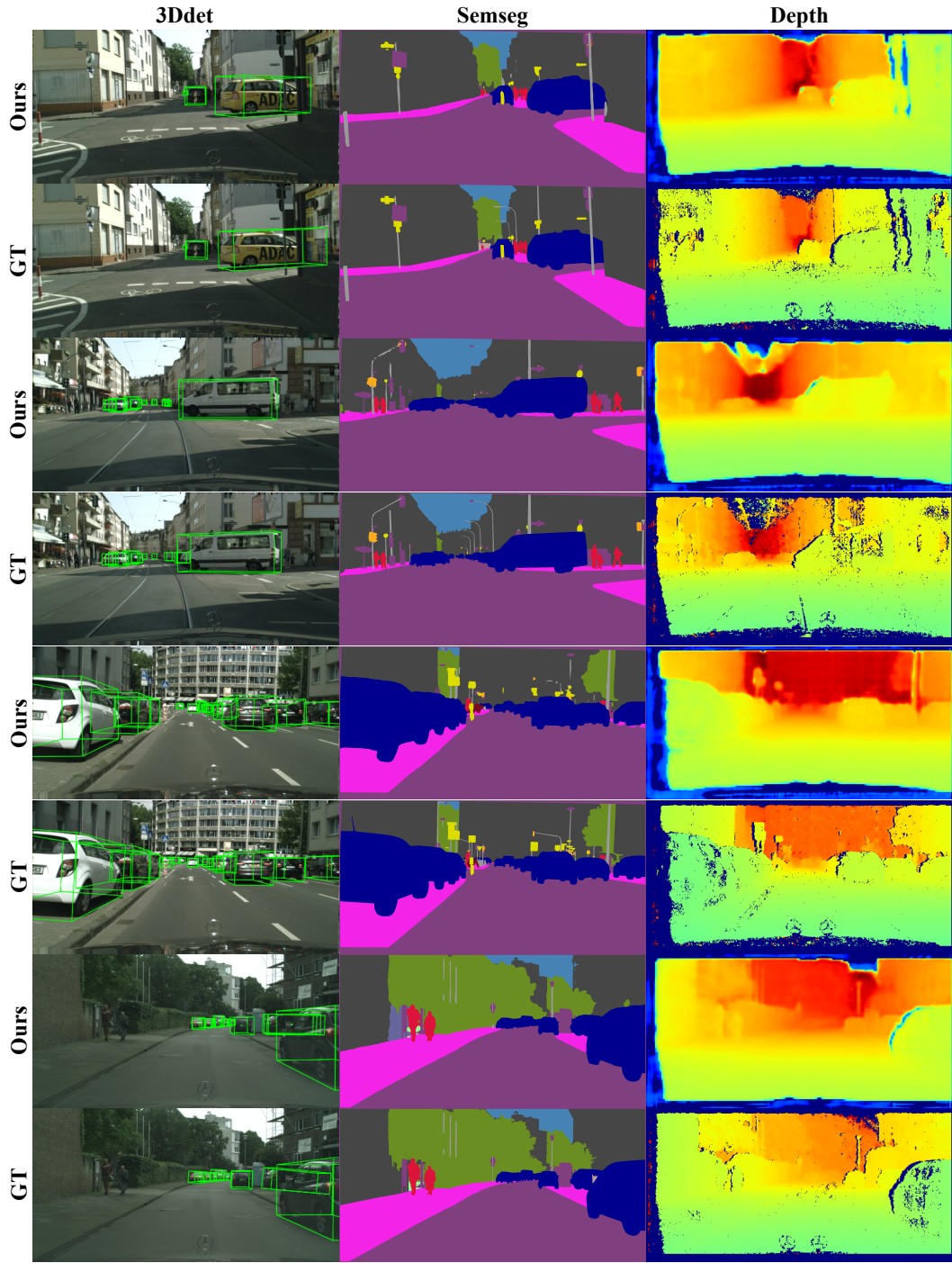

Figure 11: Prediction visualization on Cityscapes-3D. TaskPrompter can generate competitive results for multiple 2D and 3D scene understanding tasks including 3d detection, semantic segmentation, and depth estimation simultaneously.

1024×2048. For the evaluation of semantic segmentation, we use the more challenging 19-class labels. The models are evaluated on the validation set for all the tasks. 3D detection (3Ddet) uses mean detection score (mDS) as metric, which is the official evaluation index provided by Cityscapes-3D.

**Implementation of TaskPrompter on Cityscapes-3D** Since the images in Cityscapes-3D have a larger spatial resolution and 3Ddet is sensitive to object sizes, we build our multi-task prompting method on Swin-Base model (Liu et al., 2021b), which maintains high-resolution features in the backbone. Notably, as the transformer block in Swin-Base uses local window attention instead of global attention as ViT, we clone the spatial task prompts and embed them into each local window to model the interaction with all patch tokens. After computing the window attention, the spatial task prompts of different windows are merged into one by taking their average. In the decoding stage, we stitch the Spatial-Task-Prompt affinity tensors from all windows as a global affinity tensor to query the patch tokens. The channel task prompt learning and decoding are not affected by the Swin-Base architecture, which can be performed in the same way as in ViT backbone. The channel number of task prompts is doubled at each Patch Merging layer of Swin-Base to maintain the same channel number as patch tokens. The same as those used in experiments on PASCAL-Context and NYUD-v2, we use task-specific 3x3Conv-BN-ReLU blocks as prediction heads for generating final task features for all 3 tasks, and a linear layer for generating the final predictions for Semseg and Depth. As for 3Ddet, we adopt the final prediction heads of FCOS-3D (Wang et al., 2021) to predict the location coordinates, rotation angles, sizes, object classes, center-ness, and direction classification. To reduce computation cost, we reduce the resolution of images from $1024\times2048$ to $768\times1536$. The batch size is set to 2 and the model is trained for 40k iterations. We also use Adam optimizer with a learning rate $2 \times 10^{-5}$ without weight decay. For the 3D detection task, Non-maximum Suppression is used with a threshold of 0.3.

**Loss Functions** For 3D detection, similar to FCOS3D (Wang et al., 2021), we use focal loss (Lin et al., 2017) for object classification, smooth $\mathcal{L}1$ loss for location coordinates and size regression, and cross-entropy loss for direction classification and center-ness regression.

Table 4: Performance of joint 2D-3D multi-task scene understanding on Cityscapes-3D dataset. TaskPrompter achieves better or comparable results against SOTA methods of multiple tasks. Bold denotes the best.

| Model | | 3Ddet mDS ↑ | Semseg mIoU ↑ | Depth RMSE ↓ |
|---|---|---|---|---|
| Single-task Models | **3Ddet**: One-Stage (Haq et al., 2022) | 26.90 | - | - |
| | **Semseg**: SETR w/ ViT-B (Zheng et al., 2021) | - | **78.02** | - |
| | **Depth**: SDC-Depth (Wang et al., 2020) | - | - | 6.92 |
| Multi-task Models | Our Baseline | 29.69 | 76.90 | 7.00 |
| | **TaskPrompter** | **32.94** | 77.72 | **6.78** |

**Experimental Results on Cityscapes-3D** The overall performance comparison with state-of-the-art methods (single-task models) and multi-task baseline on Cityscapes-3D is shown in Table 4. The multi-task baseline adopts a similar design as the multi-task baseline on PASCAL-Context but is built upon Swin-base backbone. It should be noted that we are the first in the literature to simultaneously perform all three tasks on this dataset. We clearly observe that our TaskPrompter can significantly improve our baseline on all three tasks, further confirming the effectiveness of the method. Moreover, TaskPrompter even yields better performance than several SOTA single-task models, such as on 3Ddet (Haq et al., 2022) and Depth (Wang et al., 2020). It also shows decent performance on the highly competitive task (*e.g.* Semseg) compared with the SOTA (Zheng et al., 2021). We also visualize our prediction results on Cityscapes-3D and compare them with ground truth labels in Fig. 11. TaskPrompter can generate competitive results for multiple 2D and 3D scene understanding tasks simultaneously. These experiment results further indicate that the proposed TaskPrompter can be effectively adapted to other transformer models and task sets.

## A.7 COMPARISON WITH SOTA METHODS USING ViT-LARGE BACKBONE

As transformer-based multi-task learning methods only appear recently (Ye & Xu, 2022), we re-implement several CNN-based SOTA methods, including ATRC (Bruggemann et al., 2021), MTI-Net (Vandenhende et al., 2020), and PAD-Net (Xu et al., 2018), on ViT-Large backbone. We compare the performances of TaskPrompter and these methods in Table 5. We observe that our TaskPrompter outperforms all previous SOTA methods with less computation cost. The reason is that TaskPrompter avoids using a heavy multi-task decoder as previous decoder-focused methods

do and propose a novel Dense Spatial-Channel Task Prompt Decoding to decode multi-task dense predictions with the help of task prompts.

Table 5: Comparison with the state-of-the-art methods on PASCAL-Context. Our TaskPrompter achieves clearly superior performances on *all tasks* with the same backbone.

| Model | FLOPs | #Param | Semseg mIoU ↑ | Parsing mIoU ↑ | Saliency maxF ↑ | Normal mErr ↓ | Boundary odsF ↑ |
|---|---|---|---|---|---|---|---|
| PAD-Net (Xu et al., 2018) | 124G | 81M | 53.60 | 59.60 | 65.80 | 15.30 | 72.50 |
| MTI-Net (Vandenhende et al., 2020) | 161G | 128M | 61.70 | 60.18 | 84.78 | 14.23 | 70.80 |
| ATRC (Bruggemann et al., 2021) | 216G | 96M | 67.67 | 62.93 | 82.29 | 14.24 | 72.42 |
| PAD-Net w/ ViT-L (Xu et al., 2018) | 773G | 330M | 78.01 | 67.12 | 79.21 | 14.37 | 72.60 |
| MTI-Net w/ ViT-L (Vandenhende et al., 2020) | 774G | 851M | 78.31 | 67.40 | 84.75 | 14.67 | 73.00 |
| ATRC w/ ViT-L (Bruggemann et al., 2021) | 871G | 340M | 77.11 | 66.84 | 81.20 | 14.23 | 72.10 |
| InvPT (Ye & Xu, 2022) | 669G | 423M | 79.03 | 67.61 | 84.81 | 14.15 | 73.00 |
| **TaskPrompter (ours)** | 497G | 401M | **80.89** | **68.89** | **84.83** | **13.72** | **73.50** |

## A.8 Ablation Study of Using Task-Specific Encoders

To verify the importance of the joint learning of task-specific, task-generic, and cross-task interaction in TaskPrompter, we design a model variant of TaskPrompter that uses task-specific encoder for each task. We name this variant "TaskPrompter w/ TE". In each task-specific encoder, we use one task prompt for the corresponding task, and thus there is no task-generic feature in the encoder stage. We still use our Dense Spatial-Channel Task Prompt Decoding to generate prediction for each task from the task-specific feature of the encoder. We adopt ViT-S as model backbone and compare the model variant to TaskPrompter with also ViT-S backbone in Table 6. We find that when using task-specific only encoders, the model performance decreases on all tasks, despite the model capacity and computation cost being much larger as each task has an independent encoder. The results can verify the importance of jointly modeling both task-specific and task-generic features in TaskPrompter, in terms of both effectiveness and efficiency.

Table 6: Ablation study of using task-specific encoders (TE) with ViT-S backbone. '↓' means lower better and '↑' means higher better.

| Model | FLOPs | #Param | Semseg mIoU ↑ | Parsing mIoU ↑ | Saliency maxF ↑ | Normal mErr ↓ | Boundary odsF ↑ |
|---|---|---|---|---|---|---|---|
| TaskPrompter | 205G | 132M | 76.57 | 63.22 | 84.54 | 13.93 | 70.60 |
| TaskPrompter w/ TE | 400G | 495M | 74.47 | 61.99 | 84.33 | 14.15 | 70.40 |

Table 7: More ablation study of the effectiveness of different components of TaskPrompter.

| Model | Semseg mIoU ↑ | Parsing mIoU ↑ | Saliency maxF ↑ | Normal mErr ↓ | Boundary odsF ↑ | MTL Gain $\Delta_m$ ↑ |
|---|---|---|---|---|---|---|
| TaskPrompter | 79.00 | 67.00 | 85.05 | 13.47 | 73.50 | 0.15 |
| - CPrompt | 78.10 | 65.99 | 84.88 | 13.58 | 72.10 | -0.96 |
| - RW | 78.55 | 66.02 | 84.22 | 13.60 | 71.80 | -1.10 |

## A.9 More Ablation Studies of Different Modules Combinations

To further examine the influence of different combinations of modules, we investigate another two combinations of modules for ablation study. In the first variant, we remove CPrompt, *i.e.*, channel task prompt learning, from TaskPrompter with ViT-B backbone. In the second variant, we remove RW, *i.e.*, cross-task reweighting. The results are shown in Table 7. As we can observe from the results, if we disable CPrompt and RW from the TaskPrompter respectively, the model performance shows a clear decrease on all five tasks in both cases, further confirming the effectiveness of channel task prompt learning and the cross-task reweighting strategies.

## A.10 Model Efficiency with Different Numbers of Tasks

TaskPrompter is efficient compared with the previous transformer-based method with heavy decoder design (Ye & Xu, 2022). The reason is that TaskPrompter utilizes task prompts to help decode task-specific features by querying the image patch-tokens through task-prompt-to-patch affinities without

Table 8: Study of the efficiency of TaskPrompter with different numbers of tasks. The last column reports the increases in the computation costs of the model from one task to five tasks.

| # Task | 1 | 2 | 3 | 4 | 5 | Increase |
|---|---|---|---|---|---|---|
| # Param | 364M | 373M | 383M | 392M | 401M | +10.16% |
| FLOPs | 389G | 416G | 443G | 470G | 497G | +27.76% |

using an additional decoder. We show the number of parameters and FLOPs of TaskPrompter with different numbers of tasks on PASCAL-Context in Table 8. We observe that from 1 task to 5 tasks we only increase 10.16% parameters and 27.76% FLOPs, which demonstrates a strong scaling ability of our multi-task model.

