# OpenReview forum: "TaskPrompter: Spatial-Channel Multi-Task Prompting for Dense Scene Understanding"
_ICLR.cc/2023/Conference — ICLR 2023 poster_

### Official Review · Reviewer_GABz · 2022-10-19

**Confidence:** 4
**Correctness:** 3
**Technical Novelty And Significance:** 3
**Empirical Novelty And Significance:** 2
**Recommendation:** 6

**Clarity, Quality, Novelty And Reproducibility:**

I have stated the clarity and quality in the previous section. I believe the architecture itself is novel. But the architecture design lacks proper motivation and definitely requires some further clarification.

**Strength And Weaknesses:**

Strength.

- The experiments were conducted showing consistently improved performance over a range of competitive baselines.
- The visualisation of per-task affinity map looks interesting and intuitive, showing the effectiveness of the proposed design.

Weakness.

- Motivation and Design. The paper mentions in multiple places claiming that prior methods such as Cross-Stitch Netwoks, MTAN, and MTI-Net learn cross-task information through hand-craft structures, with task general, task shared representations were structured in each individual modules. However, from my perspective, the proposed method also learns these representations in different modules. Specifically, we may argue that the TaskPromper even further splits task-specific representation into spatial and channel modules, with cross-task information, are encoded in the decoder layers. As such, the proposed design does not strictly follow the motivation.
- Prompt or Tricks? In Table 1, we can find each design component contributes some sense of performance improvements. Among which, the hierarchical prompting (which is a common design strategy in decoder-based MTL) contributes the most. This also makes me concerned that the overall performance is mainly because of the hierarchical design rather than the complicated prompting module.
- Why channel? I did not follow the motivation of proposing channel prompts in the spatial-channel task prompt learning module design. Why learning interactions among different channels is important for MTL? Sure I can find the improved performance in Table 1, but this does not justify the design. Overall, I found learning channel interactions is very strange and unintuitive. It may be possible that the improved performance is mainly from the additional cross-attention layer.
- Lack of Efficiency measurement. One important aspect of MTL architecture design is efficiency. As highlighted in multiple baseline methods such as MTI-Net, PAD-Net, it's important to show the size of network in terms of number of tasks. This allows us to understand the computation cost of the proposed design with the scaling of different number of tasks.


**Summary Of The Paper:**

This paper proposes a multi-task architecture based on transformers for dense prediction tasks, named TaskPrompter. The design of TaskPromper was motivated by learning task-general, task-specific and cross-task relationships all into a unified learning module. To achieve this, the paper designs a spatial-channel prompting technique that jointly learns spatial and channel interactions through two self-attention layers. Additionally combined with other design strategies such as hierarchical prompting and cross-task reweigthing, the proposed architecture reaches a great performance on a range of multi-task benchmarks.

**Summary Of The Review:**

The paper proposes a multi-task architecture TaskPrompter for dense pixel-wise scene understanding. The architecture itself is complicated in design, though achieving great performance in standard multi-task benchmarks. The study of the contribution of each design component is limited, and lack of motivation. I am currently holding a borderline score for this work, and will raise/lower the score depending on further rebuttal.

---

> ### Public Comment · ~Du_Can1 · 2022-11-07
> **Experimental comparisons are NOT FAIR**
>
> Weakness:
>
> 1 Competitive baseline selection is not equivalent.  ViT-L backbone v.s HRNet backbone is not fair. This manuscript uses big Tranformer-based backbone (ViT-L). Most comparisons use HRNet-18/48 as backbone in Tab. 3.
>
> 2 Missing important GFLOPs and Parameters. The previous ATRC published the GFLOPs and Parameters on the NYUD-v2 dataset.

---

> > ### Author Response · Authors · 2022-11-19
> > **Response to Public Commenter "Du Can"**
> >
> > Please see the response [here](https://openreview.net/forum?id=-CwPopPJda&noteId=rbYPSvm-Oz), thank you.

---

> ### Author Response · Authors · 2022-11-19
> **Response to GABz (1/2)**
>
> We thank the reviewer for the positive feedback and insightful comments. We address the detailed questions and comments below.
>
> >R5.Q1. From my perspective, the proposed method also learns these representations in different modules. Specifically, we may argue that the TaskPromper even further splits task-specific representation into spatial and channel modules, with cross-task information, are encoded in the decoder layers. As such, the proposed design does not strictly follow the motivation.
>
> It is a challenge in multi-task learning to have an effective design to simultaneously learn task-generic representations, task-specific representations, and cross-task interactions. TaskPrompt enables the learning of these three types of information in each layer of the transformer encoder: (i) The global task prompts are embedded into the transformer encoder from the first layer, which model global task-specific representations; (ii) The image patch tokens learned from the input image are shared for all the tasks, which are task-generic representations; (iii) The cross-task interactions are modeled among the task prompts and the image patch tokens via long-range self-attention. These three types of information are thus *jointly* learned by self-attention in the proposed unified Spatial-Channel Task Prompt Learning Module. To learn more representative task prompts, the task prompts are further projected into spatial and channel task prompts to have interactions with the image patch tokens from both the spatial and channel dimensions, as the spatial and channel information is critically important for dense prediction problems. The spatial-channel interaction is still conducted in the same Spatial-Channel Task Prompt Learning module in the same transformer layer, as shown in Fig. 2.
>
> Regarding existing state-of-the-art multi-task works, such as Cross-Stitch Networks, MTAN, MTI-Net, ATRC, InvPT, they all learn task-generic features and task-specific features in different modules. More specifically, in Cross-Stitch, each task uses an independent backbone network, meaning that there is no parameter sharing for the different tasks. Thus, it only models task-specific features. They design a separate cross-stitch module to combine features from different backbone networks to model the cross-task interactions. As each task requires an independent network, the structure is not scalable to a relatively large number of tasks, and in their paper, only two tasks are considered. Regarding MTAN, MTI-Net, ATRC, and InvPT, they clearly have a shared backbone for learning task-generic features, and then have carefully designed separate decoders to respectively learn task-specific features. None of these works proposes a design that can jointly model these three types of information (i.e., task-generic, task-specific representations, and cross-task interactions) in one unified module.
>
> >R5.Q2. Concern that the overall performance is mainly because of the hierarchical design instead of the prompting module.
>
> Thank the reviewer for the comment. We would like to point out that the hierarchical prompting is also a part of our proposed multi-task prompting module. Because it is essentially the spatial-channel hierarchical prompting, which is directly operated based on our learned spatial and channel task prompts at each layer of the transformer encoder. With the embedding and learning of the global task prompts from the beginning of the transformer encoder, our model can naturally perform spatial-channel hierarchical prompting at each layer for task-specific features.
>
> Besides, even if we perform prompting for task-specific features only at the last layer of the transformer encoder, we also achieved clear improvements over the baseline on all the five tasks, with an overall MTL gain of 2.21 points, as shown in Table 2. On the other hand, previous methods also adopt information from different layers of the backbone, such as PAD-Net, MTI-Net, and InvPT, and our method also clearly outperforms theirs with the same backbone, as shown in Table 3 of the paper and Table 5 in Appendix.

---

> > ### Comment · Reviewer_GABz · 2022-11-26
> > **Response to the Rebuttal**
> >
> > Thanks to the authors for providing a detailed and comprehensive rebuttal supported with additional experiments.
> >
> > My concerns have been resolved. And after checking with other reviewers'comments, I would keep my weak acceptance as my final rating, conditioned on these additional analyses and experiments will be included in the final paper.

---

> ### Author Response · Authors · 2022-11-19
> **Response to GABz (2/2)**
>
>
>
> >R5.Q3. Why learning interactions among different channels is important for MTL?
>
>
> The benefits of using both spatial and channel information for feature refinement have been widely demonstrated in various computer vision tasks, such as image classification (e.g. SE-Net [2]) and dense prediction (e.g. dual attention network [3]). In TaskPrompter, the learning of the task prompts is through the interactions between the task prompts and the image patch tokens. To benefit largely from both the spatial and channel information from image patch tokens, we thus propose to project task prompts into spatial task prompts and channel task prompts to have dense interactions with image patch tokens from these two complementary dimensions, to learn more representative task prompts. Our ablation study in Table 2 also clearly verifies the effectiveness of jointly modeling spatial-temporal interactions with the image patch tokens for multi-task prompt learning. Besides, as shown in Fig. 4, the qualitative visualization of the channel-task-prompt affinities can also confirm that the different tasks show different responses to different channels of the image patch tokens.
>
>
> >R5.Q4. It's important to show the size of network in terms of number of tasks. This allows us to understand the computation cost of the proposed design with the scaling of the different numbers of tasks.
>
>
> Thank the reviewer for the suggestion. We show the number of parameters and FLOPs of TaskPrompter with different numbers of tasks in the table below. It can be observed that from 1 task to 5 tasks, TaskPrompter only increases 10.16% parameters and 27.76% FLOPs, which demonstrates a strong scaling ability of our TaskPrompter with respect to the number of tasks. The reason is that the TaskPrompter designs task prompts to decode task-specific features by querying the image patch-tokens through task-prompt-to-patch affinities, without requiring an additional decoder. We have added this efficiency analysis and discussion in Appendix A.11.
> |# Task |  1 |  2 |  3 |  4 |  5 |  Cost Increase from 1 task to 5 tasks |
>  | --- | --- | --- | --- | --- | --- | --- |
>  |# Parameters  | 364M | 373M | 383M | 392M | 401M | +10.16% |
>  | FLOPs  | 389G | 416G | 443G | 470G | 497G | +27.76% |
>
>
> >R5.Q5. Further explanation of architecture design.
>
> Thank the reviewer for the comments. The proposed multi-task spatial-channel prompting method is shown in the framework overview in Fig. 2, which we believe is concise and enables joint modeling of task-generic, task-specific representations, and cross-task interactions at each layer of the transformer encoder. This design can decode task-specific features directly using the learned spatial and channel task prompts (which are task-specific tokens) to query the task-generic image tokens at different layers, without requiring specially designed heavy task decoders as in previous state-of-the-art works, such as PAD-Net, MTI-Net, and InvPT,  thus simplifying the overall multi-task dense prediction framework. We also implement it as a unified network module, which can be flexibly plugged into any transformer layer to replace the vanilla self-attention module for multi-task learning. We conduct an extensive investigation to demonstrate the effectiveness of the different components, as shown in Table 1 in the paper. The overall results of our method significantly surpass the previous CNN-based methods, and also outperform the best-performing transformer-based pipeline in the literature.
>
> [1] Vandenhende et al, MTI-Net: Multi-Scale Task Interaction Networks for Multi-Task Learning, ECCV 2020.
>
> [2] Hu et al., Squeeze-and-Excitation Networks, CVPR 2018.
>
> [3] Fu et al., Dual Attention Network for Scene Segmentation, CVPR 2019.

---

### Official Review · Reviewer_NUwM · 2022-10-24

**Confidence:** 4
**Correctness:** 3
**Technical Novelty And Significance:** 3
**Empirical Novelty And Significance:** 2
**Recommendation:** 6

**Clarity, Quality, Novelty And Reproducibility:**

The model design is new and interesting. It seems helpful to improve spatial awareness for the transformer.

**Strength And Weaknesses:**

Strengths:
1. The paper is well-written and easy to follow. The authors presented details of their module designs.
2. The proposed idea is interesting and technically valid for dense scene prediction tasks.
3. The authors conducted several analyses to verify the importance of the proposed components.
4. The experiments are thorough and clear.

Weaknesses:
1. The paper only considered multi-task scene-based dense prediction datasets. I wonder whether the authors consider the generic benchmark, such as ADE20K, COCO-stuff and Cityscapes. I understand the authors applied it to multi-task problem to demonstrate the cross-task interaction/contribution, however, since the authors claimed it is able to learn task-specific representations, it will be more convincing to compare with those SOTA on ADE20K and others.
2. In Table 2, adding hierarchical prompting seems the most effective one compared to others. The authors may want to discuss more about the reasons, but not just talk about the empirical observations in appendix.
3. In Table 2, I wonder if the model in each row has the same model size? If the model sizes are different, how to verify the performance improvements are attributed to the proposed components or extra parameters?
4. While the model design is good and interesting, the model seems complex and heavy compared to existing frameworks. Would it be possible to provide some discussions regarding model speed and complexity?


**Summary Of The Paper:**

This paper presents a transformer framework with spatial-channel multi-task prompting for both generic and task-specific feature learning.

The authors observe that both spatial and channel information are important for visual dense prediction tasks. Thus, the authors propose to use spatial-channel task prompts to learn their relationships with image tokens. This design could better train the transformer to capture useful pixel information for better feature learning.

While the prompts are usually task-specific, the authors also found that different prompts can contribute to cross-task performance enhancement.

Experiments on NYUD-V2 and Pascal-Context are strong.

**Summary Of The Review:**

The authors proposed an interesting transformer architecture design that could be useful for dense prediction tasks. However, the architecture design is a lot more complex and not easy to reproduce.

While the authors focused on multi-task learning and shown good signals in cross-task interaction learning, it is unclear how the proposed model performs compared to single-task SOTA on the well-known benchmarks as mentioned in Weaknesses.  Since the authors claimed their model has ability for the learning of task-generic and task-specific representations, additional comparisons might be added. The paper may need some clarifications or adjustments. I will update my rating based on the authors' feedback.

---

> ### Author Response · Authors · 2022-11-19
> **Response to Reviewer NUwM (1/2)**
>
> Thank the reviewer for the constructive feedback. We address the detailed questions and comments below.
>
> >R4.Q1. The paper only considered multi-task scene-based dense prediction datasets. I wonder whether the authors consider the generic benchmark, such as ADE20K, COCO-stuff and Cityscapes.
>
> This paper targets the multi-task dense prediction problem. Following previous methods, such as InvPT (ECCV2022),  ATRC (ICCV2021), and MTI-Net (ECCV2020), we evaluated TaskPrompter on three challenging and well-established benchmarks for the problem, i.e., PASCAL-Context, NYUD-v2, and Cityscapes.
>
> However, there are no existing results for multi-task learning that can be compared with previous works on ADE20K and COCO-stuff, which are commonly used for single-task learning, typically for semantic segmentation or object detection. One reason why PASCAL-Context and NYUD-v2 datasets are widely used for multi-task learning is that, these two datasets provide more numbers of (e.g., pascal-context offering 5 different tasks) and more diverse (containing both 2D and 3D tasks, e.g. depth, surface normal, and semantic) tasks for the evaluation of multi-task learning methods. It is particularly important that a multi-task model can show robust multi-task performance on a relatively large number of distinct and diverse tasks, which is a challenge of multi-task learning.
>
> Moreover, in this paper, to further examine the performance of TaskPrompter, we indeed conducted experiments on Cityscapes for multi-task learning shown in Appendix A.6, together with the paper submission. We set up a challenging 2D-3D multi-task joint learning problem on Cityscapes, which simultaneously involves 3D object detection, monocular depth estimation, and semantic segmentation. Our TaskPrompter yields very competitive performances on the different 2D and 3D perception tasks compared with the very strong single-task state-of-the-art methods on this dataset, further demonstrating the effectiveness, and the generalization capability to different sets of tasks of the proposed model. Some qualitative results on Cityscapes are also shown in Fig. 11 in Appendix.
>
>
> >R4.Q2. More Analysis of Hierarchical Prompting (HP)
>
> Thank the reviewer for the suggestion. The hierarchical prompting is essentially the spatial-channel hierarchical prompting, which is directly operated based on our learned spatial and channel task prompts at each layer of the transformer encoder. With the embedding and learning of the global task prompts from the beginning of the transformer encoder, our model can naturally perform spatial-channel hierarchical prompting at different layers for decoding task-specific features. We add the related analysis in "Effectiveness of Cross-Task Reweighting and Hierarchical Prompting" of Section 4.2.
>
> Besides, even if we perform prompting for task-specific features only at the last layer of the transformer encoder, we also achieved clear improvements over the baseline on all five different tasks, with an overall MTL gain of 2.21 points, as shown in Table 2. On the other hand, previous methods also adopt information from different layers of the backbone, such as PAD-Net, MTI-Net, and InvPT, and our method also clearly outperforms theirs with the same backbone, as shown in Table 3 of the paper and Table 5 in Appendix.
>
>
> >R4.Q3. In Table 2, I wonder if the model in each row has the same model size? If the model sizes are different, how to verify the performance improvements are attributed to the proposed components or extra parameters?
>
> Thanks to the reviewer for the comments. We think the reviewer is mentioning Table 1 in the main paper. To prove that the effectiveness of TaskPrompter does not result from simple parameter increase, we design a model variant based on “Baseline” of Table 1 by adding a MTI-Net [1] multi-task decoder, and show the FLOPs and number of parameters as well as performances of different models with ViT-B backbone in the table below. We find that although MTI-Net requires more parameters and computation cost, its performance is clearly worse than our TaskPrompter, which suggests that adding model capacity doesn’t necessarily lead to better performance, and our TaskPrompter brings solid performance improvement.
>
> | Model  |FLOPs | Params  | Semseg (mIOU) |Parsing (mIOU) |Saliency (maxF) |Normal (mErr) |Boundary (odsF) |
>  | --- | --- | --- | --- | --- | --- |  --- |  --- |
>   | Baseline + MTI-Net Decoder | 519G  | 633M | 76.50 | 66.70 | 84.67 | 14.68 | 72.60
>  |TaskPrompter |420G | 353M  |  79.00 |  67.00  | 85.05  | 13.47 |  73.50

---

> > ### Comment · Reviewer_NUwM · 2022-11-28
> > **Responds**
> >
> > Thank you for the detailed explanations and additional experiments. Most of my concerns have been adequately addressed. I will update the rating as weak accept.

---

> ### Author Response · Authors · 2022-11-19
> **Response to Reviewer NUwM (2/2)**
>
>
> >R4.Q4. While the model design is good and interesting, the model seems complex and heavy compared to existing frameworks. Would it be possible to provide some discussions regarding model speed and complexity?
>
> Thank the reviewer for the suggestion. The proposed spatial-channel task prompt learning module is shown in the framework overview in Fig. 2, which we believe is concise and enables joint modeling of task-generic, task-specific representations, and cross-task interactions. This design can decode task-specific features directly using the learned spatial and channel task prompts, without requiring specially designed heavy task decoders, thus simplifying the overall multi-task dense prediction framework. We also implement it as a unified module, which can be flexibly plugged into any transformer layer to replace the vanilla self-attention module for multi-task learning.
>
> As suggested by the reviewer, we also show the FLOPs and the number of parameters of TaskPrompter in the table below. TaskPrompter achieves higher efficiency compared to the best-performing transformer-based method in the literature (i.e., InvPT [2]), which employs the same transformer backbone model as TaskPrompter. The main reason is that InvPT considers a heavy decoder structure to perform multi-task feature learning, while our model can directly decode task-specific features from the encoder using the proposed multi-task prompting method. We add more efficiency studies in Appendix A.7.
>
> Model   | FLOPs   | Params   |
>  | --- | --- | --- |
> InvPT (ECCV2022)   | 669G  |  423M   |
> TaskPrompter (ours)  |  497G  |  401M   |
>
> >R4.Q5. Implementation and Reproducibility
>
> For the reproducibility of TaskPrompter, as promised in the paper abstract, we will open-source our code and models, to benefit the community. TaskPrompter is an elegant framework to perform effective multi-task representation learning, based on the designed spatial-channel task prompting method. It does not require complicated designs of different task decoders, as in many existing works. More implementation details of TaskPrompter have also been illustrated in Appendix A.3 due to the page limit.
>
> >R4.Q6. Since the authors claimed their model has the ability for the learning of task-generic and task-specific representations, additional comparisons might be added.
>
> Task-generic and task-specific representations are widely used concepts in multi-task learning [3]. Task-generic representations indicate the features shared by all tasks, while task-specific representations indicate the features used to decode each specific task. They are different by their definitions, and existing multi-task models typically learn both types of representations for better multi-task prediction. However, in contrast to existing models, our proposal can jointly learn task-generic features, task-specific features, and cross-task interactions at each layer of the transformer with the proposed multi-task spatial-channel prompting strategy. Besides the quantitative ablation studies that show the effectiveness of our design, as shown in Fig. 4 of the main paper, the qualitative results also confirm that TaskPrompter can effectively learn task-specific affinity maps for different tasks, which are used to produce task-specific features for multi-task predictions.
>
> [1]  Vandenhende et al, MTI-Net: Multi-Scale Task Interaction Networks for Multi-Task Learning, ECCV 2020.
>
> [2] Ye et al, Inverted Pyramid Multi-task Transformer for Dense Scene Understanding, ECCV2022.
>
> [3] Vandenhende et al, Multi-Task Learning for Dense Prediction Tasks: A Survey, TPAMI, 2021.

---

### Official Review · Reviewer_Xzsy · 2022-10-24

**Confidence:** 3
**Correctness:** 4
**Technical Novelty And Significance:** 3
**Empirical Novelty And Significance:** 3
**Recommendation:** 8

**Clarity, Quality, Novelty And Reproducibility:**

The paper is clear, well-written, and of high quality. The design idea of using "task prompts", which are task-specific learnable tokens to learn spatial and channel-wise task-specific information for each task, although it has been introduced before, for other NLP tasks, their application for dense prediction in visual tasks is original.

**Strength And Weaknesses:**

Strengths:
* The topic is of great interest to the research community
* The method has a high degree of novelty
* Good ablation studies
* Method agnostic of the different types of transformer architectures

Weaknesses:
* Minor typos throughout the manuscript - I recommend proof-reading once again
* There is a minor error in the Appendix - Table 4 - the authors should bold the number from SETR-vitB (Zheng et al., 2021) since their performance is better than the one from TaskPrompter.


**Summary Of The Paper:**

The paper introduces TaskPrompter - a multi-task transformer network for dense scene understanding. The framework has three parts - (1) Prompt Embedding, (2) Spatial-Channel Task Prompt Learning, and (3) Dense Spatial-Channel Task Prompt Decoding - with a strong focus on the latter two - which constitutes the main contributions of the paper. The effectiveness of the method and its components is thoroughly proven through multiple ablation studies. Experimental analysis is performed on popular benchmarks for multi-task learning - Pascal-Context and NYUD-v2 (the appendix also provides state-of-the-art results on Cityscapes-3D on two of the three tackled tasks).

**Summary Of The Review:**

The paper has no significant flaws, the method is sound, and it has a good amount of novelty. The experimental setup is fair with convincing results over previous state-of-the-art methods. The reviewer has no grounds for rejecting this paper.

---

> ### Public Comment · ~Du_Can1 · 2022-11-07
> **Experimental comparisions are UNFAIR**
>
> Weaknesses:
>
> 1 Ablation study is NOT adequate. The authors miss the important ablation using CNN-based backbone.
>
> 2 Missing the ablation about the number layer of stacks of the proposed Spatial-Channel Task Prompt module.
>
> 3 Comparisons are NOT FAIR. In Tab.3, six of the seven comparison methods are based on HRNet-18/48 backbone. However, the authors use the ViT-L as backbone. The big backbone (ViT-L) can bring a high level of improvement, NOT the method itself.
>
> 4 Missing important GFLOPs and Parameters.  The authors are advised to follow the ATRC setting instead of InvPT.

---

> > ### Author Response · Authors · 2022-11-19
> > **Response to Public Commenter "Du Can"**
> >
> > Please see the response [here](https://openreview.net/forum?id=-CwPopPJda&noteId=rbYPSvm-Oz), thank you.

---

> ### Author Response · Authors · 2022-11-19
> **Response to Reviewer Xzsy**
>
> We would like to thank the reviewer for the valuable suggestions. We have fixed the bold issue in the Appendix in our revision. We also conducted careful proof-reading again as suggested.

---

### Official Review · Reviewer_y2MQ · 2022-10-30

**Confidence:** 3
**Correctness:** 3
**Technical Novelty And Significance:** 3
**Empirical Novelty And Significance:** 3
**Recommendation:** 6

**Clarity, Quality, Novelty And Reproducibility:**

Clarity: overall good
Quality: good
Novelty: good
Reproducibility: should be possible to reproduce

**Strength And Weaknesses:**

### Strength
* The proposed method can put most of the network's capacity on learning all tasks simultaneously; the cross-task representation interactions are also modeled.
* The novelty is good. I feel it's quite new to involve the prompting concept into multi-task vision problems.
* Experiments show that the model's performance is good.

### Weaknesses
* It seems that the model and experiments assume that supervision for all tasks are there. It should be easy to extend the work on dataset where each example may have supervision of a subset of the tasks. This may be a more practical and flexible setting, as during the inference time, user may only need a single task for an example.
* The design of "Spatial-Channel Task Prompt Learning Module" is a bit complicated. I am wondering if it makes sense to make the model an encoder-decoder design, where the encoder encodes the images, and the decoder handles the prompts and cross-attend to the image embeddings. Also it would be necessary to analysis the computation overhead of the proposed method.

**Summary Of The Paper:**

This paper address the problem of multi-task dense scene understanding, including semantic segmentation, human parsing etc. A transformer based method is proposed, and task-specific prompt tokens are designed to enable the transformer architecture to utilize its full capacity on all tasks. Authors modified the multi-head attention module by a spatial-channel task prompt learning module, and the decoding part is separately designed to produce predictions. Experiments are carried out on several scene understanding datasets to demonstrate its efficacy.

**Summary Of The Review:**

The paper is interesting and novel, and I feel it may be helpful for future vision model design.

---

> ### Public Comment · ~Du_Can1 · 2022-11-07
> **Experimental comparisons are NOT FAIR**
>
> Weaknesses:
>
> 1 Comparisons are NOT FAIR. This manuscript uses big Tranformer-based backbone  (ViT-L). Most comparisons use HRNet-18/48 as backbone in Tab. 3.
>
> 2 Missing important GFLOPs and Parameters. The previous ATRC published the GFLOPs and Parameters on the NYUD-v2 dataset.
>
> 3 Missing the ablation about the number layer of stacks of the proposed Spatial-Channel Task Prompt module.

---

> > ### Author Response · Authors · 2022-11-19
> > **Response to Public Commenter "Du Can"**
> >
> > Please see the response [here](https://openreview.net/forum?id=-CwPopPJda&noteId=rbYPSvm-Oz), thank you.

---

> ### Author Response · Authors · 2022-11-19
> **Response to Reviewer y2MQ**
>
> We thank the reviewer for the positive feedback and insightful comments. We address the detailed questions and comments below.
>
> >R2.Q1. It seems that the model and experiments assume that supervision for all tasks are there. It should be easy to extend the work on dataset where each example may have supervision of a subset of the tasks.
>
> Thank the reviewer for pointing out the potential of the proposed method for partially supervised multi-task dense prediction, although TaskPrompter is designed to target the problem of supervised multi-task dense prediction, using well-established benchmarks for the evaluation of the method.
>
> Following the suggestion of the reviewer, we perform an experiment by only using partial-task annotations to train our framework, which is actually a partially-supervised setting for multi-task dense prediction recently proposed in [1]. Our framework is indeed flexible for learning with only partial task annotations, since our framework models global task and data interactions and consistency in each transformer layer by self-attention, which can help unlabeled samples obtain supervision from labeled samples and is beneficial for the framework to learn with only partial annotations. We strictly follow the one-label setup designed in [1] to perform the experiment. One-label setup indicates that each image sample in train split is only associated with label for one task. The performance is shown in the table below. We can observe that Taskprompter with partial supervision (one-label) is clearly worse than that of using all the task labels for each sample, while the performance is still quite promising, which confirms the advantage of our model, and triggering a new potential direction of using our multi-task prompting framework for the problem of partially supervised multi-task dense prediction. We have added this interesting result and analysis in the Appendix A.10.
>
> | Model     | Semseg (mIOU) |Parsing (mIOU) |Saliency (maxF) |Normal (mErr) |Boundary (odsF) |
>  | --- | --- | --- | --- | --- | --- |
> TaskPrompter w/ full supervision  | 79.00 |  67.00 |  85.05 |  13.47 |  73.50 |
> TaskPrompter w/ partial supervision (one-label as in [1])  | 77.47  | 65.22 |  84.96 |  14.19 |  71.30 |
>
> >R2.Q2. I am wondering if it makes sense to make the model an encoder-decoder design, where the encoder encodes the images, and the decoder handles the prompts and cross-attend to the image embeddings.
>
> We thank the reviewer for the comment. Our multi-task prompting method can also be used in the decoder if we employ an encoder-decoder structure since the multi-task prompting method is implemented as a unified module to replace the vanilla self-attention module in any transformer layer. We do not consider it because of two reasons: (i) we want to avoid using a heavy decoder structure for learning task-specific features, which may significantly increase the network overhead as multiple tasks have to be modeled simultaneously with separate decoders. Based on our designed multi-task prompting module, the task-generic, task-specific representations, and the cross-task interactions can be jointly learned in the same module at each layer of the transformer encoder. (ii) directly embedding the global task prompts into the encoder instead of decoder can largely benefit from pretrained parameters, which can facilitate the learning of the task prompts, and the optimization of the overall framework, which is also an advantage of prompting schemes demonstrated in other NLP problems.
>
> >R2.Q3.  It would be necessary to analyze the computation overhead of the proposed method.
>
> Thank the reviewer for the suggestion. We show the FLOPs and the number of parameters of TaskPrompter in the table below. TaskPrompter achieves higher efficiency compared to the best-performing transformer-based method in the literature (i.e., InvPT [2]), which employs the same transformer backbone model as TaskPrompter. The main reason is that InvPT[2] considers a heavy decoder structure to perform multi-task feature learning, while our model directly decodes task-specific features from the encoder using the proposed multi-task spatial-channel prompting method. We add more efficiency study in Appendix A.7.
>
> Model   | FLOPs   | Params   |
>  | --- | --- | --- |
> InvPT (ECCV2022)   | 669G  |  423M   |
> TaskPrompter (ours)  |  497G  |  401M   |
>
> [1] Li et al, Learning Multiple Dense Prediction Tasks from Partially Annotated Data, CVPR2022.
>
> [2] Ye et al, Inverted Pyramid Multi-task Transformer for Dense Scene Understanding, ECCV2022.

---

### Official Review · Reviewer_CiXq · 2022-11-01

**Confidence:** 3
**Correctness:** 3
**Technical Novelty And Significance:** 3
**Empirical Novelty And Significance:** Not applicable
**Recommendation:** 8

**Clarity, Quality, Novelty And Reproducibility:**

The overall architecture and approach presented is novel. Due to the complex nature of the architecture, the description of the method is not always clear and thus reproducibility could be difficult.

**Strength And Weaknesses:**

Strength
- The overall architecture. Developing an encoder and decoder under such settings is challenging and the author’s were able to include task-specific and task-generic representation learning. The architecture can be applied to many different tasks.
- The idea of using prompts to represent the different tasks

Weaknesses
- The main weakness is that it is unclear from the experiments how important it is to have task-specific and task-generic features in both the encoder and decoder. This could be addressed through additional experiments where a task-specific only encoder is used.
- The experimental section could use more details. For example, what is the ‘Task Prompter Baseline?’ Hierarchical Prompting seems to have the largest increase in performance. What happens when different combinations of modules are added?

**Summary Of The Paper:**

TaskPrompter presents a new framework for multi-task dense scene understanding. The framework assigns each task a  learnable token which allows task-specific and task-generic learning in both the encoder and decoder. The task tokens and image tokens interact through the attention layers in transformers.


**Summary Of The Review:**

I recommend acceptance because of the novelty of the architecture: the lack of task-specific components and using prompting to represent different tasks.

---

> ### Public Comment · ~Du_Can1 · 2022-11-07
> **Experimental comparisons are NOT FAIR**
>
> Weaknesses:
>
> 1 Comparisons are NOT FAIR. In Tab.3,  six of the seven comparison methods are based on HRNet-18/48 backbone. However, the authors use the ViT-L as backbone. The big backbone (ViT-L) can bring a high level of improvement, NOT the method itself. Therefore, the authors should add HRNet-18/48 as backbone experiments.
>
> 2 Missing important GFLOPs and Parameters. They can be very objective to evaluate the efficiency of the model. The ATRC published the GFLOPs and Parameters on the NYUD-v2 dataset.  The authors are advised to follow the ATRC setting instead of InvPT.

---

> > ### Author Response · Authors · 2022-11-19
> > **Response to Public Commenter "Du Can"**
> >
> > Please see the response [here](https://openreview.net/forum?id=-CwPopPJda&noteId=rbYPSvm-Oz), thank you.

---

> ### Author Response · Authors · 2022-11-18
> **Response to Reviewer CiXq**
>
> We thank the reviewer for the positive feedback and insightful comments. We address the questions and comments below.
>
> > R1.Q1. It is not clear from the experiments how important it is to have task-specific and task-generic features in both the encoder and decoder.  This could be addressed through additional experiments where a task-specific only encoder is used.
>
> We want to clarify that the proposed methods are only used at each layer of the transformer encoder to simplify the whole framework, which embeds global task prompts at the beginning of the encoder to learn task-generic, task-specific representations, and cross-task interactions in each encoder layer. We then use the learned global task prompts to query the task-generic image patch tokens, which directly decodes task-specific features for multi-task prediction. Therefore, our method does not require a heavy design for task-specific decoders. The details for decoding task-specific features with the global spatial-channel task prompts are illustrated in Section 3.3, i.e., Dense Spatial-Channel Task Prompt Decoding.
>
> As suggested by the reviewer, we add an experiment of using task-specific only encoders in TaskPrompter with ViT-S backbone. In each task-specific encoder, we use one task prompt for the corresponding task, and thus there is no task-generic feature in the encoder stage. We still use our Dense Spatial-Channel Task Prompt Decoding to generate prediction for each task from the task-specific feature of the encoder. We compare this variant to our TaskPrompter with also ViT-S backbone in the table below. We find that when using task-specific only encoders, the model performance decreases on all tasks, despite the model capacity being much larger since each task has an independent encoder. The results can verify the importance of jointly modeling both task-specific and task-generic features in TaskPrompter, in terms of both effectiveness and efficiency. We have added this experiment in Appendix A.8.
>
> | Model |FLOPs | Params  | Semseg (mIOU) |Parsing (mIOU) |Saliency (maxF) |Normal (mErr) |Boundary (odsF) |
>  | --- | --- | --- | --- | --- | --- | --- | --- |
> | TaskPrompter ViT-S  | 205G| 132M  |76.57 | 63.22 | 84.54 | 13.93 | 70.60
> | TaskPrompter ViT-S w/ task-specific only encoders  |  400G | 495M |   74.47  |61.99  | 84.33  | 14.15  | 70.40  |
>
> >R1.Q2. Experimental section could use more details, for example, the model baseline ‘Task Prompter’.
>
> We defined all the details of different model variants in Section 4.1 ( i.e., Models Declaration). Specifically, the TaskPrompter Baseline is built upon a ViT-Base with 12 transformer layers, which is pre-trained on ImageNet-22K, and uses task-specific prediction heads, i.e., Conv (3x3)-BN-ReLU-Conv(1x1) blocks, for final multi-task predictions. We have revised this part to make it more clear.
>
> >R1.Q3. What happens when different combinations of modules are added?
>
> As suggested by the reviewer, we conduct two additional ablation studies for TaskPrompter. We investigate another two combinations of modules, which are SPrompt+RW+HP (remove CPrompt, i.e.,  channel task prompt learning) and SPrompt+CPrompt+HP (remove RW, i.e., cross-task reweighting) to further show the effect of CPrompt and RW. The results are shown in the table below. As we can observe from the results, if we disable CPrompt and RW from the TaskPrompter respectively, the model performance shows a clear decrease on all five different tasks under both cases, further confirming the effectiveness of the CPrompt and the RW strategies.  We have added these results in Appendix A.9.
>
> | Model     | Semseg (mIOU) |Parsing (mIOU) |Saliency (maxF) |Normal (mErr) |Boundary (odsF) | MTL Gain |
>  | --- | --- | --- | --- | --- | --- |  --- |
>  | +SPrompt + CPrompt + RW + HP |  79.00 |  67.00  | 85.05  | 13.47 |  73.50 | 0.15
>  | +SPrompt+RW+HP (remove CPrompt) |  78.10 | 65.99 | 84.88 | 13.58 | 72.10 | -0.96
>  | +SPrompt+CPrompt+HP (remove RW)  | 78.55  | 66.02 |  84.22  | 13.60  | 71.80 | -1.10
>
> >R1.Q4. Reproducibility
>
> For the reproducibility of TaskPrompter, as promised in the paper abstract, we will open-source our code and models, to benefit the community. TaskPrompter is an elegant framework to perform effective multi-task representation learning, based on the designed spatial-channel multi-task prompting method. It does not require complicated designs of different task decoders, as in many existing works. More implementation details of TaskPrompter have also been illustrated in Appendix A.3 due to the page limit.

---

> > ### Comment · Reviewer_CiXq · 2022-11-26
> > **Updated correctness**
> >
> > Thank you to the authors for the updated results. I rate the correctness as a 4 instead of 3 based on these results.

---

### Author Response · Authors · 2022-11-19
**Response to Public Commenter "Du Can"**

Thanks for the comments. Since you have repeatedly posted similar comments under different reviewers, we think it is better to reply in an independent comment box to avoid verbosity.

>Q1 The big backbone (ViT-L) can bring a high level of improvement. The authors should add HRNet-18/48 as backbone experiments.

The effectiveness of TaskPrompter has been clearly demonstrated in Table 1, where all the model variants are based on the same backbone, and we observe effective performance improvement brought by our proposal.

Regarding the SOTA comparison in Table 3, the most competitive, and the most recent SOTA work we compare is InvPT [1]. We use the same backbone as theirs, and on all tasks of PASCAL-Context and NYUD-v2, our TaskPrompter surpasses InvPT on all 9 tasks on these two popular benchmarks, which, again, strongly suggests the effectiveness of the design of TaskPrompter. Moreover, our method achieves higher efficiency compared to InvPT in terms of the number of parameters and the FLOPs, as can be seen in the efficiency analysis and discussion in Appendix A.7.

As suggested by this commenter, to further compare the performance of TaskPrompter with previous CNN-based methods, we reimplement the backbone of ATRC [2], MTI-Net [3], and PAD-Net [4] with ViT-L (the same backbone as ours), and show the FLOPs, the number of parameters, and the performances in the table below. We can observe that our TaskPrompter still achieves the best results, and requires even lower computational cost because TaskPrompter avoids the usage of a heavy multi-task decoder.

| Model  |FLOPs | Params  | Semseg (mIOU) |Parsing (mIOU) |Saliency (maxF) |Normal (mErr) |Boundary (odsF) |
 | --- | --- | --- | --- | --- | --- |  --- |  --- |
  | PAD-Net w/ ViT-L |  773G  | 330M  | 78.01  | 67.12  | 79.21  | 14.37  | 72.60  |
 | MTI-Net w/ ViT-L  | 774G |  851M  | 78.31  | 67.40  | 84.75 | 14.67  | 73.00  |
 | ATRC w/ ViT-L|  871G  | 340M  | 77.11  | 66.84  | 81.20  | 14.23  | 72.10  |
 | InvPT w/ ViT-L  | 669G  | 423M  | 79.03  | 67.61  | 84.81  | 14.15  | 73.00  |
 | TaskPrompter (ours)  | 497G | 401M  | 80.89  | 68.89  | 84.83  | 13.72  | 73.50 |

>Q2 Why not use CNNs as backbone as some previous methods?

The choice of using transformers instead of CNNs as the backbone is mainly due to two reasons:

(i) Transformer-based architectures have shown significant advantages in global task and data modeling compared to local CNN models, and have achieved remarkably better results compared to CNN-based pipelines;

(ii) Prompt tuning mechanism is designed based on transformers [5,6,7], which is not straightforward to explore them in CNN models, and the motivation of using global prompting for local CNNs is weird. Our TaskPrompter focuses on designing an effective multi-task prompting method to jointly learn task-generic, task-specific, and cross-task interactions in a unified layer, which can produce discriminative multi-task representations without requiring complex task-specific decoders.

Besides, our ablation study has clearly demonstrated the effectiveness of our multi-task prompting method by using the same backbone architecture for the baseline and the different variants.


>Q3 Ablation study about the number of layers stacks with Spatial-Channel Task Prompt Learning Modules

We believe it is a misunderstanding about our method. We do not stack the proposed Spatial-Channel Task Prompt Learning Modules in the transformer. Instead, as clearly presented in Fig. 2 of the paper, TaskPrompter replaces the vanilla self-attention module of each layer of the transformer encoder (e.g., ViT) with the proposed Spatial-Channel Task Prompt Learning module, to jointly learn task-generic, task-specific representations, and cross-task interactions at each layer of the transformer encoder.

[1] Ye et al, Inverted Pyramid Multi-task Transformer for Dense Scene Understanding, ECCV2022.

[2] Bruggemann et al, Exploring Relational Context for Multi-Task Dense Prediction, ICCV2021.

[3] Vandenhende et al, MTI-Net: Multi-Scale Task Interaction Networks for Multi-Task Learning, ECCV2020.

[4] Xu et al, PAD-Net: Multi-Tasks Guided Prediction-and-Distillation Network for Simultaneous Depth Estimation and Scene Parsing, CVPR2018.

[5] Jia et al, Visual Prompt Tuning, ECCV2022.

[6] Li et al, Prefix-Tuning: Optimizing Continuous Prompts for Generation, ACL 2021.

[7] Liu et al, Pre-train, Prompt, and Predict: A Systematic Survey of Prompting Methods in Natural Language Processing, arXiv 2021.

---

### Decision · Program_Chairs · 2023-01-20

**Decision:**

Accept: poster

**Justification For Why Not Higher Score:**

The current recommendation is based on the reviewers’ scores. A higher score seems to be not well justified.

**Justification For Why Not Lower Score:**

All reviewers recommend acceptance. I agree with this recommendation. Authors should attend to main points in the reviews, especially additional analyses and experiments, when preparing a final version. Also, as promised by the authors, the codebase should be open sourced.

**Metareview: Summary, Strengths And Weaknesses:**

This paper proposes a transformer-based multi-task architecture called TaskPrompter for dense visual prediction tasks. By assigning each task a learnable prompt token, TaskPrompter enables task-specific and task-agnostic learning in both the encoder and decoder, and also encourages cross-task representation interactions via attention layers with spatial-channel prompting. Hierarchical prompting and cross-task reweighting are further leveraged. Extensive experiments along with ablation studies validate the effectiveness of TaskPrompter and its components.

Generally, the reviewers consider the contribution of introducing the prompting concept (or learnable tokens) that have been widely-used in other fields like NLP into the dense visual prediction tasks as novel and solid. The approach seems to be general, e.g., with respect to different supervision settings and different types of transformer architectures. The performance improvements are noticeable and consistent. In the original manuscript, some implementation and experimental details, design rationale, effect of different components (especially the hierarchical prompting), and analysis on model complexity and efficiency were missing or unclear. The authors’ responses together with additional experimental results have successfully addressed the reviewers’ concerns as well as the public comments.

All reviewers recommend acceptance. I support the reviewers’ recommendation. Authors should attend to main points in the reviews, especially additional analyses and experiments, when preparing a final version. Also, as promised by the authors, the codebase should be open sourced.

**Note From Pc:**

if the above contains the word "oral" or "spotlight" please see: "oral" presentation means -> notable-top-5% and "spotlight" means -> notable-top-25%. As stated in our emails, we are disassociating presentation type from AC recommendations